# Aligning climate scenarios to emissions inventories shifts global benchmarks

Matthew J. Gidden[1,2,10 ✉], Thomas Gasser[1,10], Giacomo Grassi[3], Nicklas Forsell[1], Iris Janssens[1,4], William F. Lamb[5,6], Jan Minx[5,6], Zebedee Nicholls[1,7,8], Jan Steinhauser[1,9] & Keywan Riahi[1]

Taking stock of global progress towards achieving the Paris Agreement requires consistently measuring aggregate national actions and pledges against modelled mitigation pathways[1]. However, national greenhouse gas inventories (NGHGIs) and scientific assessments of anthropogenic emissions follow different accounting conventions for land-based carbon fluxes resulting in a large difference in the present emission estimates[2,3], a gap that will evolve over time. Using state-of-the-art methodologies[4] and a land carbon-cycle emulator[5], we align the Intergovernmental Panel on Climate Change (IPCC)-assessed mitigation pathways with the NGHGIs to make a comparison. We find that the key global mitigation benchmarks become harder to achieve when calculated using the NGHGI conventions, requiring both earlier net-zero $CO_2$ timing and lower cumulative emissions. Furthermore, weakening natural carbon removal processes such as carbon fertilization can mask anthropogenic land-based removal efforts, with the result that land-based carbon fluxes in NGHGIs may ultimately become sources of emissions by 2100. Our results are important for the Global Stocktake[6], suggesting that nations will need to increase the collective ambition of their climate targets to remain consistent with the global temperature goals.

The 2021 UN Climate Change Conference (COP26) marked a shift in the focus of climate policy from pledge-making to implementation towards the long-term temperature goal of the Paris Agreement, the collective progress towards which is assessed through periodic Global Stocktakes (GSTs). In spring 2022, the first GST was launched[7] and continues through the 2023 United Nations Climate Change Conference (COP28) to establish evaluation mechanisms among parties. Comparing present emission trends from the NGHGIs and future targets in a collective benchmarking effort rooted in the best available science will be key for a rigorous, precedent-setting first GST and the overall success of the Paris Agreement[6].

Countries have gradually increased the ambition of their national targets in response to the latest IPCC report findings[1,8]. Notably, several nations made long-term net-zero emission commitments in the run-up to COP26 (ref. 9), which brought the long-term temperature goal of the Paris Agreement within striking distance, although much of the assessed temperature reductions arose from long-term and non-binding promises rather than immediate climate action[10–12]. Global climate scenarios show that both deep reductions of near-term emissions as well as enhancement of anthropogenic land-based carbon sinks are needed to achieve net-zero emissions and limit global warming to achieve the temperature goal of the Paris Agreement[13,14].

A key discrepancy exists, however, in how model-based scientific studies and NGHGIs account for the role of anthropogenic land-based carbon fluxes[4,15,16], with national inventories incorporating a broader scope of removals[2], resulting in lower emission estimates in NGHGIs. Previous studies[2–4] have quantified the magnitude of this difference to be approximately 5.5–6.7 Gt $CO_2$ $yr^{-1}$. This conceptual difference hinders the comparability of the aggregate targets set by countries and future mitigation benchmarks. Although this problem has been acknowledged in the most recent IPCC assessment[17] and raised by parties during the GST[18], the impact of this discrepancy on national and global mitigation benchmarks is still not well understood. Aligning mitigation pathways assessed by the IPCC with NGHGI conventions is therefore needed to support the science-based formulation of nationally determined contributions (NDCs) and to measure collective global action against emission levels necessary to achieve the Paris Agreement goal.

## Aligning climate pathways and inventories

The IPCC-assessed mitigation pathways are typically generated by integrated assessment models (IAMs) that capture transitions in anthropogenic energy and land-use systems consistent with stated global climate policy objectives. The reporting conventions for land-use, land-use change and forestry (LULUCF) carbon fluxes of these models follow that of detailed global carbon-cycle models (that is, 'bookkeeping' models). These models simulate and account for direct anthropogenic fluxes (due to human-induced land-use changes, forest harvest and regrowth) separately from indirect fluxes that are the natural response

[1]International Institute for Applied Systems Analysis, Laxenburg, Austria. [2]Climate Analytics, Berlin, Germany. [3]Joint Research Centre, European Commission, Ispra, Italy. [4]Department of Computer Science, imec, University of Antwerp, Antwerp, Belgium. [5]Mercator Research Institute on Global Commons and Climate Change, Berlin, Germany. [6]Priestley International Centre of Climate, School of Earth and Environment, University of Leeds, Leeds, UK. [7]Melbourne Climate Future's Doctoral Academy, School of Geography, Earth and Atmospheric Sciences, University of Melbourne, Parkville, Victoria, Australia. [8]Climate Resource, Northcote, Victoria, Australia. [9]Potsdam Institute for Climate Impact Research, Potsdam, Germany. [10]These authors contributed equally: Matthew J. Gidden, Thomas Gasser. ✉e-mail: gidden@iiasa.ac.at

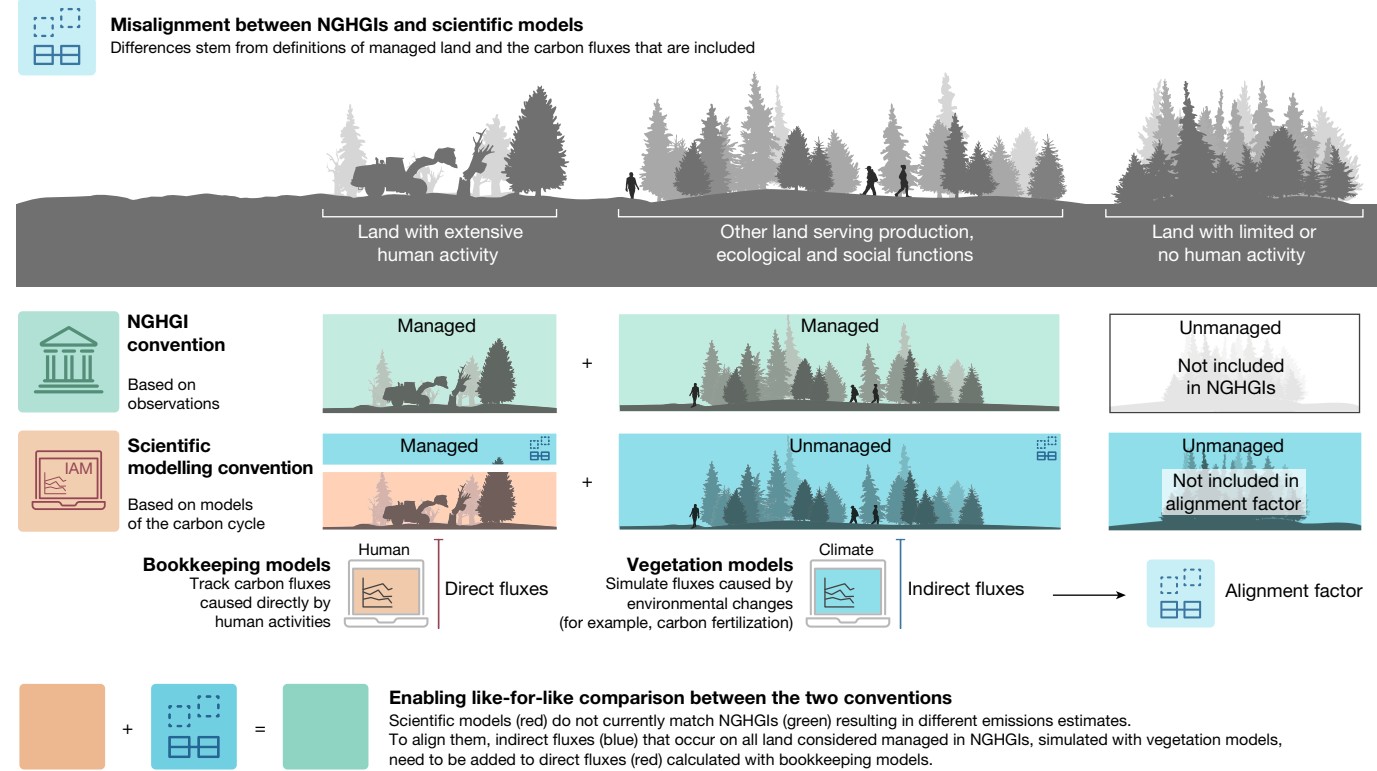

**Misalignment between NGHGIs and scientific models**
Differences stem from definitions of managed land and the carbon fluxes that are included

Land with extensive human activity | Other land serving production, ecological and social functions | Land with limited or no human activity

**NGHGI convention**
Based on observations

Managed + Managed | Unmanaged Not included in NGHGIs

**Scientific modelling convention**
Based on models of the carbon cycle

Managed + Managed | Unmanaged | Unmanaged Not included in alignment factor

**Bookkeeping models** Track carbon fluxes caused directly by human activities — Human — Direct fluxes

**Vegetation models** Simulate fluxes caused by environmental changes (for example, carbon fertilization) — Climate — Indirect fluxes → Alignment factor

**Enabling like-for-like comparison between the two conventions**
Scientific models (red) do not currently match NGHGIs (green) resulting in different emissions estimates.
To align them, indirect fluxes (blue) that occur on all land considered managed in NGHGIs, simulated with vegetation models, need to be added to direct fluxes (red) calculated with bookkeeping models.

**Fig. 1 | Difference in present estimates of LULUCF carbon fluxes under NGHGI and model-based accounting conventions.** Schematic showing the difference in accounting conventions between NGHGIs (green) and scientific models (bookkeeping models in red and vegetation models in blue). Models such as IAMs are based on bookkeeping approaches and consider direct fluxes due to land use (for example, wood harvest) and land-cover changes. Additional indirect fluxes due to evolving environmental conditions can be estimated by processed-based vegetation models. NGHGIs consider a wider managed land area and are generally based on physical observations, and thus include both direct and indirect fluxes. We use the term 'unmanaged' to describe land not considered managed by NGHGIs to be consistent with previous literature, but recognize that this includes land that has been managed by indigenous and traditional communities for centuries to millienia[38,39]. In this study, we estimate the alignment factor to translate between both conventions (the indirect flux considered in NGHGIs but not in models, blue). This factor will change over time based on future land-use decisions and overall mitigation efforts because of, for example, changing atmospheric $CO_2$ levels.

of land to environmental changes (for example, $CO_2$ fertilization or response to climate change)[5,15,19,20] and define anthropogenic emissions as those owing to the direct effect. Because it is practically not possible to separate direct and indirect fluxes through observations, the NGHGIs submitted by parties to the United Nations Framework Convention on Climate Change (UNFCCC) follow reporting conventions[21] that define anthropogenic fluxes using an area-based approach[22] in which all fluxes occurring on managed land are considered anthropogenic, with few exceptions to isolate natural disturbances[16,23,24]. The NGHGIs include a wider definition of managed land compared with models, which includes any forested area that 'perform[s] production, ecological, or social functions'[25] (Fig. 1). As a result, present-day LULUCF fluxes estimated with scientific modelling conventions indicate that the land sector is a net source of emissions[3], whereas the NGHGIs collectively report it as a net sink[26], resulting in fundamentally different present and future perspectives of the role of land-based fluxes.

To estimate the direct and indirect components of land-based carbon fluxes necessary to align mitigation pathways with conventions used in the NGHGIs, we use a reduced-complexity model with an explicit treatment of the land-use sector, OSCAR[5], one of the models used for the annual Global Carbon Budget[3], applied in a probabilistic setup and at a resolution of five global regions used in the IPCC assessments (Methods). We calculate a difference of $4.4 \pm 1.0$ Gt $CO_2$ yr$^{-1}$ in LULUCF emissions globally averaged over 2000–2020 between model-based (higher) and NGHGI-based (lower) accounting conventions, which is in line with the existing estimates[2,5]. We then assess the pathways with OSCAR to quantify how the difference between conventions evolves over time. A total of 914 of the 1,202 IPCC-assessed pathways provided sufficient land-use change data to enable this alignment (Extended Data Table 1; data are available at https://data.ece.iiasa.ac.at/genie/).

Across both the 1.5 °C and 2.0 °C scenarios (Fig. 2a,b; see definitions in the Methods), LULUCF emissions estimated using the NGHGI conventions show a strong increase in the total land sink until around mid-century. However, the NGHGI alignment factor (that is, the difference between the two accounting conventions; Fig. 2c) decreases over this period, nearing zero in the 2050s to 2060s for the 1.5 °C scenarios and the 2070s to 2080s for 2.0 °C scenarios. This convergence is primarily a result of the simulated stabilization and then decrease of the $CO_2$-fertilization effect as well as background climate warming reducing the overall effectiveness of the land sink[27,28], which in turn reduces the indirect removals included in NGHGIs. These dynamics lead to land-based emissions reversing their downward trend in most NGHGI-aligned scenarios by mid-century and result in the LULUCF sector becoming a net source of emissions by 2100 in about 25% of both the 1.5 °C and 2.0 °C scenarios.

## Global and regional ambition implications

More ambitious mitigation action is required to meet the global emission benchmarks derived from scenarios when assessed using the NGHGI conventions compared with model-based conventions (Extended Data Table 2 and Extended Data Fig. 1). The NGHGI-aligned pathways result in earlier net-zero $CO_2$ emissions by 1–5 years for the 1.5 °C and −1 to 7 years for the 2.0 °C scenarios (Fig. 3a). Emission

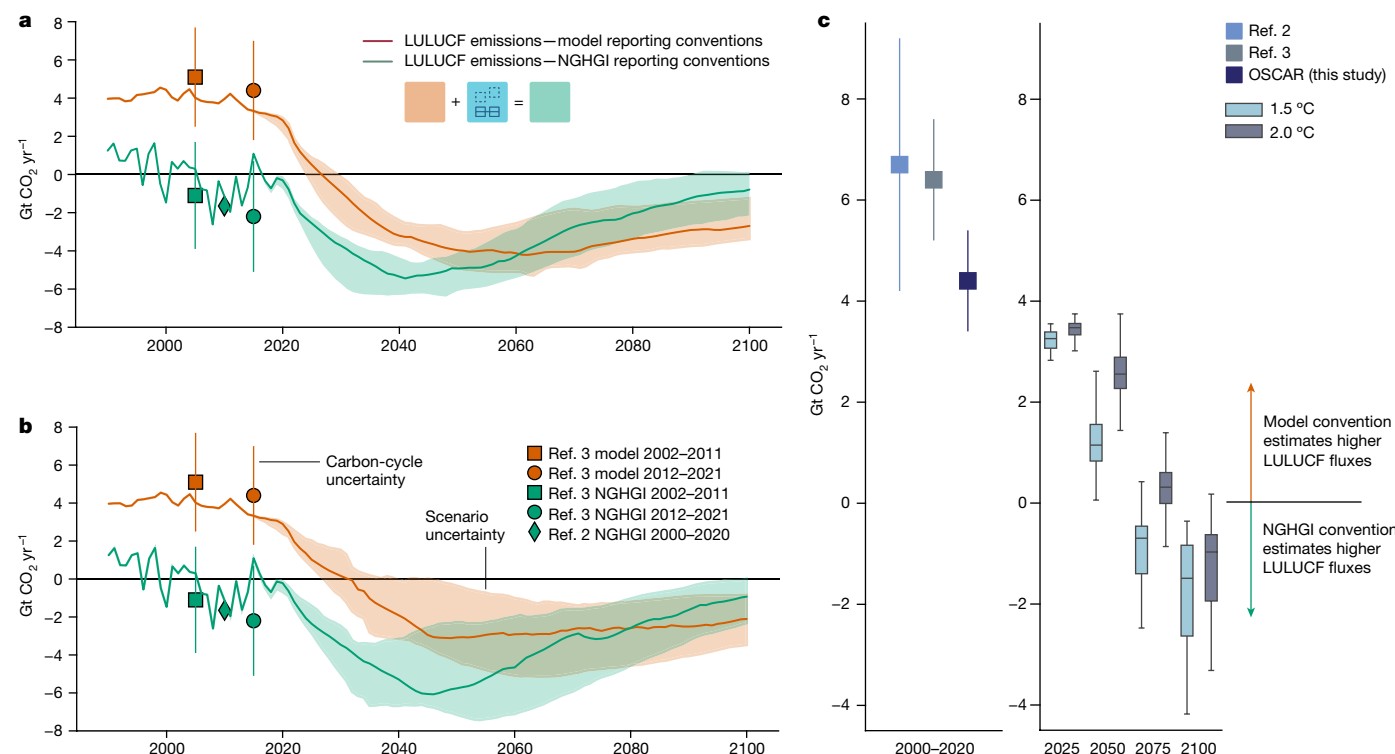

**Fig. 2 | Land-use emissions in re-analysed IPCC pathways with model-based and NGHGI-based accounting conventions. a,b,** Land-use emissions pathways before and after alignment to match NGHGIs for 1.5 °C (**a**) and 2,0 °C (**b**) pathways. Historical estimates[2,3] are shown with carbon-cycle uncertainty (1$\sigma$), and the median of scenario pathways are shown with the scenario interquartile range in shaded plumes. Pathways consistent with model-based convention are shown in red, whereas the NGHGI convention is shown in green. **c,** Comparing the two conventions results in a difference between re-analysed and NGHGI-adjusted pathways—that is, an alignment factor, which evolves as a function of the strength of land-based climate mitigation.

reductions in 2030 relative to 2020 are between 3 and 6 percentage points greater for both pathway categories (Fig. 3b). The assessed cumulative net $CO_2$ emissions to global net-zero $CO_2$ also decreases by 15–18% for both the 1.5 °C and 2.0 °C scenarios (Fig. 3c) because of extra land-based carbon removal when using the NGHGI conventions.

Although the NGHGI-aligned benchmarks strengthen, they are still consistent with the climate assessment of the IPCC. All land-use fluxes (direct and indirect) are included in the physical climate models used by the IPCC—that is, the temperature outcomes of each pathway are the same even if flux components are accounted for differently by models and inventories. When considering the additional land sink following the conventions of the NGHGIs, however, multiple dynamics interact that contribute to the revisions of the benchmarks, including the change in historical emission baseline, the enhanced anthropogenic land sink compared with what was reported by IAMs and declining sequestration from that additional sink in the future.

Parties to the UNFCCC use the net land $CO_2$ flux reported in the NGHGIs as a basis to assess compliance with their NDCs and track the progress of their long-term emission reduction strategies under the Paris Agreement[2,29,30] as with previous climate pacts[31]. Historically, NDCs have been compared with scenario-based estimates of needed emission reductions by either aligning the IPCC-assessed pathways to NGHGIs with constant offsetting methods[1] or excluding LULUCF emissions entirely[9,29]. Comparing our results with one of the most recent aggregate NDC estimates[1] (Methods and Extended Data Fig. 2), we find that the gap between unconditional NDCs and a median 2.0 °C outcome is approximately 18% larger, whereas our assessment of the gap between unconditional NDCs and a median 1.5 °C outcome is around 4% smaller (Extended Data Fig. 3). It is therefore important to incorporate a dynamic estimation of indirect fluxes when assessing national climate targets because their changing role in achieving these targets depends on domestic land-management decisions as well as the overall strength of global mitigation (Fig. 4).

Aligning pathways to inventory-based LULUCF accounting conventions can additionally affect how equitable mitigation action is understood, as around 60% of the historical NGHGI adjustment falls in Non-Annex I countries[26]. Assessed regionally, 1.5 °C-consistent emission reductions are higher for developed countries, whereas they are slightly lower in most developing regions when assessing scenario outcomes using the NGHGI-based conventions (Extended Data Fig. 4). In the 2.0 °C pathways, the NGHGI alignment results in more stringent 2020–2030 emission reductions globally compared with the unadjusted pathways, as the strength of the indirect flux continues to grow with increasing atmospheric carbon concentrations. This strengthening most directly affects regions with large forested areas such as Latin America and Russia, whereas others such as the Organisation for Economic Co-operation and Development (OECD) countries and Asia, on average, see a decrease in emission reductions. Our results span both positive and negative values across many regions, showcasing the diversity of future responses to land-sink changes and complexities when setting both equitable and ambitious climate targets based on national inventories. They also highlight the risk of over-dependence on land sinks to measure mitigation progress using national inventory conventions against ambitious climate targets.

## Considering carbon removal

In most 1.5 °C and 2.0 °C pathways, hundreds of gigatonnes of $CO_2$ are removed from the atmosphere over the course of this century, with ultimate levels dependent on the strength of near-term mitigation action[17,32,33]. Because our assessment relies on a bookkeeping model that explicitly tracks land carbon reservoirs, we are able to isolate

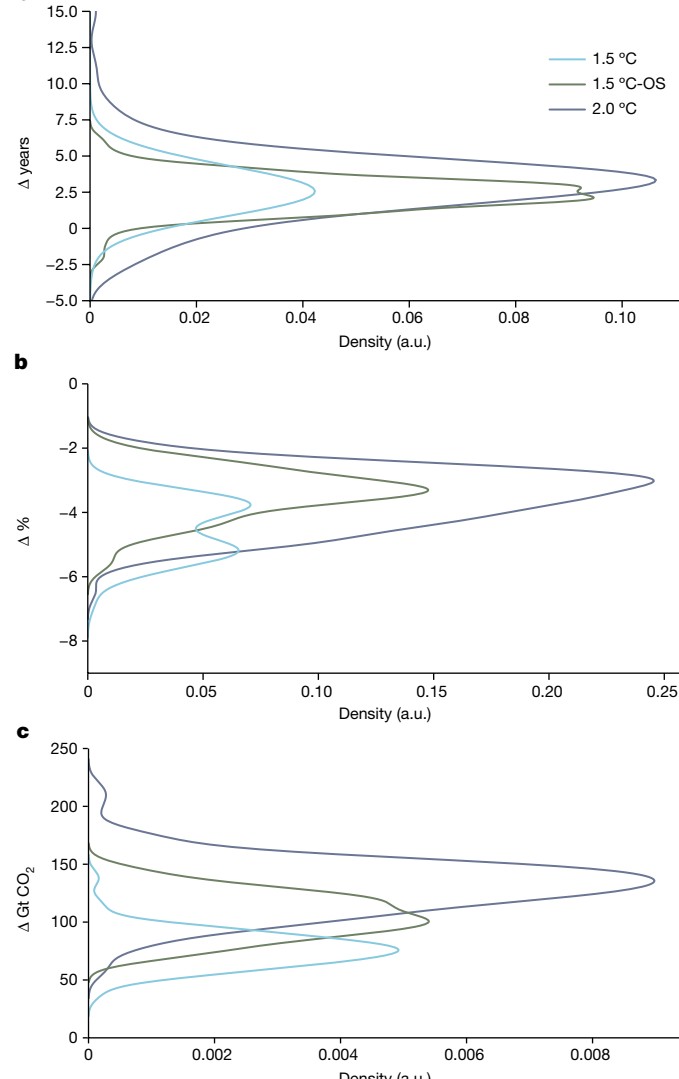

**Fig. 3 | Changes in global mitigation benchmarks across assessed scenarios. a–c,** Scenario-wise distributions of the estimated change in the net-zero $CO_2$ year (**a**), 2020–2030 $CO_2$ emission reductions (**b**) and cumulative emissions until net-zero $CO_2$ (**c**) between the re-analysed model-based and the NGHGI LULUCF accounting conventions are shown for 1.5 °C (blue, IPCC category C1), 1.5 °C-OS (green, IPCC category C2) and 2.0 °C (purple, IPCC category C3) scenarios. A positive value indicates that the benchmark comes later (for net-zero years) or is higher (for cumulative emissions) in the model-based framework compared with the NGHGI-based framework, whereas a negative value indicates that the benchmark is higher in the NGHGI-based framework (for emission reductions). Across all benchmarks, NGHGI-based accounting tends to result in more stringent outcomes (earlier net-zero years, higher emission reductions and lower cumulative emissions to net-zero $CO_2$ emission). A comparison with the original AR6 benchmarks is shown in Extended Data Fig. 1. a.u., arbitrary units.

those LULUCF fluxes that effectively constitute carbon removals from carbon emissions (for example, deforestation), thereby quantifying total land-based carbon dioxide removal (CDR) consistently across scenarios and filling a gap in the IPCC Sixth Assessment Report (see footnote 53 in ref. 17) as well as underlying scenario database that constitutes a widely used resource in the climate change community[34].

Scenarios see a marked increase by 2030 in CDR from the LULUCF sector, resulting in around 60% higher removals of $CO_2$ by 2030 compared with the 2020 levels in the 1.5 °C pathways and 10% higher removals in the 2.0 °C pathways (Fig. 5a). Taken together

with engineered (non-LULUCF) CDR options, models deploy 2.6 [1.4–3.2] Gt $CO_2$ $yr^{-1}$ (interquartile range) and 0.7 [0.3–2.5] Gt $CO_2$ $yr^{-1}$ additional CDR between 2020 and 2030 in the 1.5 °C and 2.0 °C pathways, respectively. Land-based sinks account for nearly 100% of current CDR. By 2030, in the 1.5 °C pathways, 95% [88–98%] of total CDR is delivered by land-based sinks (Fig. 5b). By 2100, CDR from LULUCF accounts for about 30% [21–42%] of the annual total.

Although deep mitigation scenarios assessed by the IPCC show a notable and continued dependence on land-based removals over the whole century, LULUCF removals of the same pathways aligned to NGHGIs would peak by mid-century and decline thereafter. Over time, the reduced effectiveness of indirect removals counterbalances the gains from direct removals[35] (Extended Data Fig. 5), maintaining the overall direct and indirect removals at around 6–7 Gt $CO_2$ $yr^{-1}$ by mid-century. The 1.5 °C pathways cumulatively sequester around 20% more carbon from direct removals but 20% less carbon from indirect removals compared with the 2.0 °C pathways over that period (Extended Data Fig. 6). Considering the changing dynamics of indirect carbon removals included in NGHGIs can dramatically change the estimated carbon removals on land over time. Although the 1.5 °C scenarios show growth in total assessed net land removal by 2030 (Fig. 5c), the scenarios aligned with current policies approximately double removals compared with the 1.5 °C and 2.0 °C scenarios by mid-century, because of the increasing strength of indirect removals (notably through strong $CO_2$ fertilization) (Fig. 5d).

Thus, although the addition of a larger 'managed land' sink in NGHGIs may reduce the reported levels of present-day national emissions in some cases, maintaining the strength of this carbon sink on these land areas may pose a fundamental challenge in the long term. Not only do estimates of needed progress in anthropogenic emission reductions risk being masked by natural sink enhancement in the near term, but even the maintenance of existing natural sinks requires additional efforts to remove carbon, for example, through the expansion of forest areas, from the NGHGI accounting perspective. In other words, the future effort needed to achieve or maintain net-zero economy-wide emissions would be underestimated using NGHGI accounting conventions as the indirect contribution to land sinks loses efficacy and may eventually become a net source of emissions in low-warming scenarios.

## Balancing practicality and policy advice

We provide here an estimation of the LULUCF emissions consistent with NGHGI accounting conventions for all IPCC-assessed scenarios that provide sufficient land-use cover information using probabilistic and constrained estimates from a single established model, OSCAR. Repeating this work with additional models would increase robustness by averaging model biases and structural uncertainties, although this would require land-use scenario information at a much finer resolution than the five regions.

Because the pathways are aligned with the NGHGI conventions by re-allocating indirect carbon fluxes caused by environmental changes to anthropogenic fluxes, our results do not change any climate outcome or mitigation benchmark produced by the IPCC, but provide a translational lens to view those outcomes from the perspective of national emissions reporting frameworks as deployed by the UNFCCC parties. For example, the fact that we find net-zero timings for the 1.5 °C pathways advance by up to 5 years compared with the IPCC-assessed benchmarks does not imply that 5 years have been lost in the race to net-zero, but that following the reporting conventions for natural sinks used by parties to the UNFCCC results in net-zero needing to be reached 5 years earlier to match the modelled benchmarks. Our results reinforce the importance of a rapid decline in fossil fuel and industry emissions in this decade while limiting reliance on nature-based solutions that can weaken over time to keep global temperature rise within the limit prescribed in the Paris Agreement.

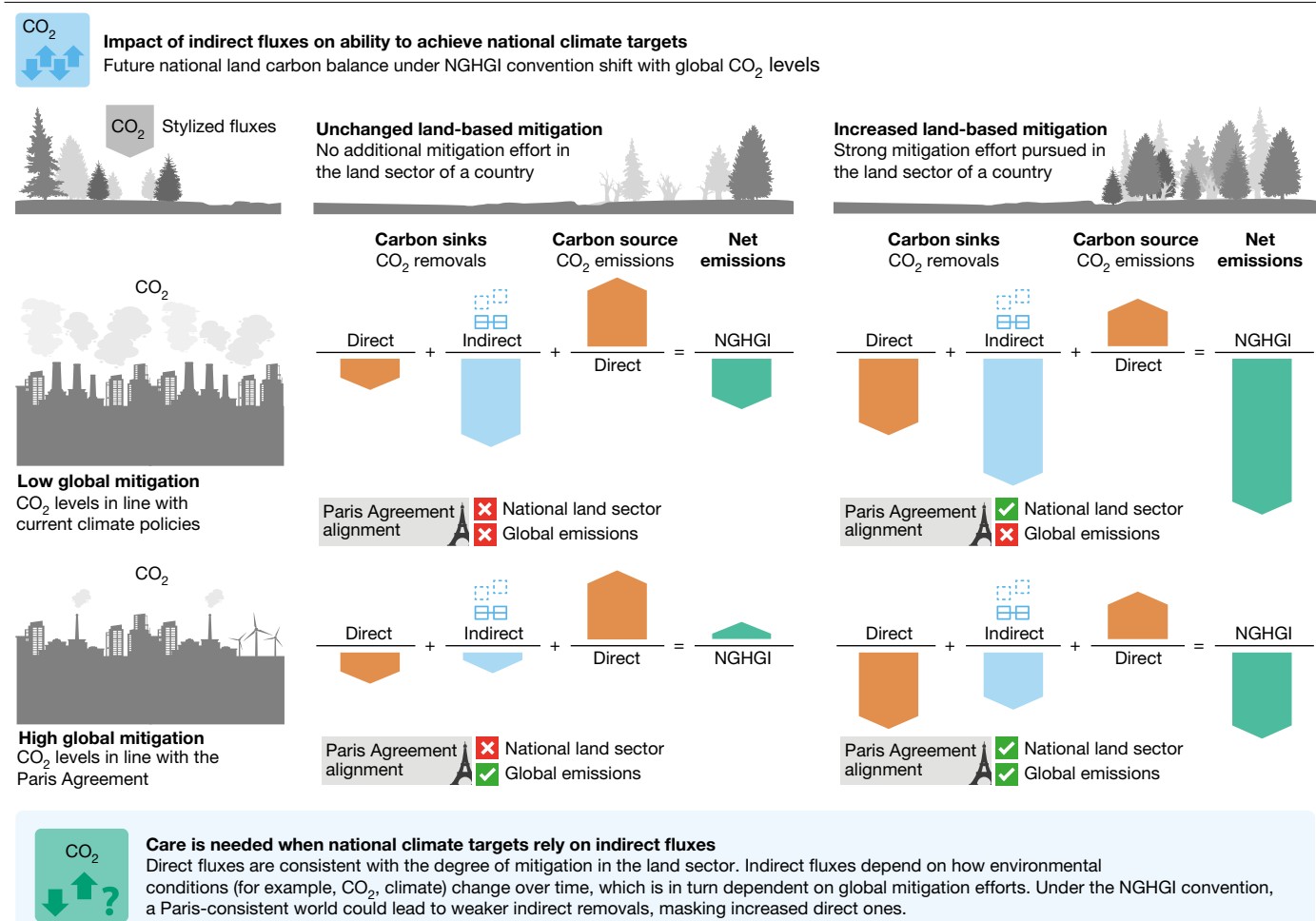

**Fig. 4 | The future role of indirect fluxes in national climate targets.** In a future with strong mitigation action in line with the goals of the Paris Agreement (bottom row), stabilizing or even decreasing atmospheric $CO_2$ will result in a weakening of the indirect sink (blue arrows), whereas a future with weak mitigation action will see the indirect sink increase (as long as $CO_2$ fertilization dominates over climate feedbacks, top row). The direct component of LULUCF fluxes (red arrows) is due entirely to land-use management decisions (columns). Future estimates of net LULUCF emissions (green arrows) will differ between conventions depending on how much overall mitigation occurs and how much land-based mitigation occurs, which can have unexpected consequences on national climate target achievement.

Importantly, this 'new' net-zero year is conceptually consistent with the meaning of balancing of sources and sinks of greenhouse gases (GHGs) as stipulated in Article 4 of the Agreement (although in the context of $CO_2$). Yet it occurs before the climatological milestone that results in halting further warming, as is the case of the net-zero year under the scientific modelling convention. Understanding and addressing how these different frameworks can be mutually interpreted is a fundamental challenge for evaluating progress towards the Paris Agreement, given the reality that carbon removals from anthropogenic and natural land-based processes cannot be estimated separately by the NGHGIs, which are typically based on direct observations. The outcomes presented here highlight that the conventions by which land-based carbon removals are considered have important implications for NDC assessment and transparency, operationalization of removals under carbon markets as laid out in Article 6.4 of the Paris Agreement and monitoring, reporting and verification of these removals.

The policy and scientific communities can take steps to meet this challenge by reconciling terms, definitions and estimates of land-based $CO_2$ fluxes in four concrete ways. First, climate targets can be formulated explicitly for areas of critical mitigation action, including gross $CO_2$ emission reductions without LULUCF, net land-based removals, engineered carbon removals and non-$CO_2$ GHG emission reductions,

allowing for parties to define their expected contributions and to measure progress in each domain separately. Second, parties can clarify the nature of their deforestation pledges, because direct and indirect carbon fluxes vary greatly in different forest types[36]. Third, scientific and practitioner communities can convene discussions on how to enhance monitoring, reporting and verification of LULUCF fluxes to better align estimates from both groups. Fourth, IAM teams can provide their individual assumptions and estimates for direct LULUCF emissions and removals, including the indirect flux component consistent with the NGHGIs[37] and their assumptions about the land-use contribution of NDCs and long-term strategies. Future IPCC assessments could either use such scenario data if available or use an approach such as that presented here to provide a holistic scenario assessment aligned with the NGHGIs and better inform necessary collective action to meet global climate goals.

Although science and policy processes continue to co-evolve, informing one another, a full reconciliation of the conceptual discrepancies outlined here will take time. However, the first iteration of the GST will be completed by the end of 2023 and new NDCs will be formulated soon thereafter, necessitating earlier compatibility between national targets and benchmarks estimated by global models. Our results provide estimates and a line of evidence that can be directly used by parties and

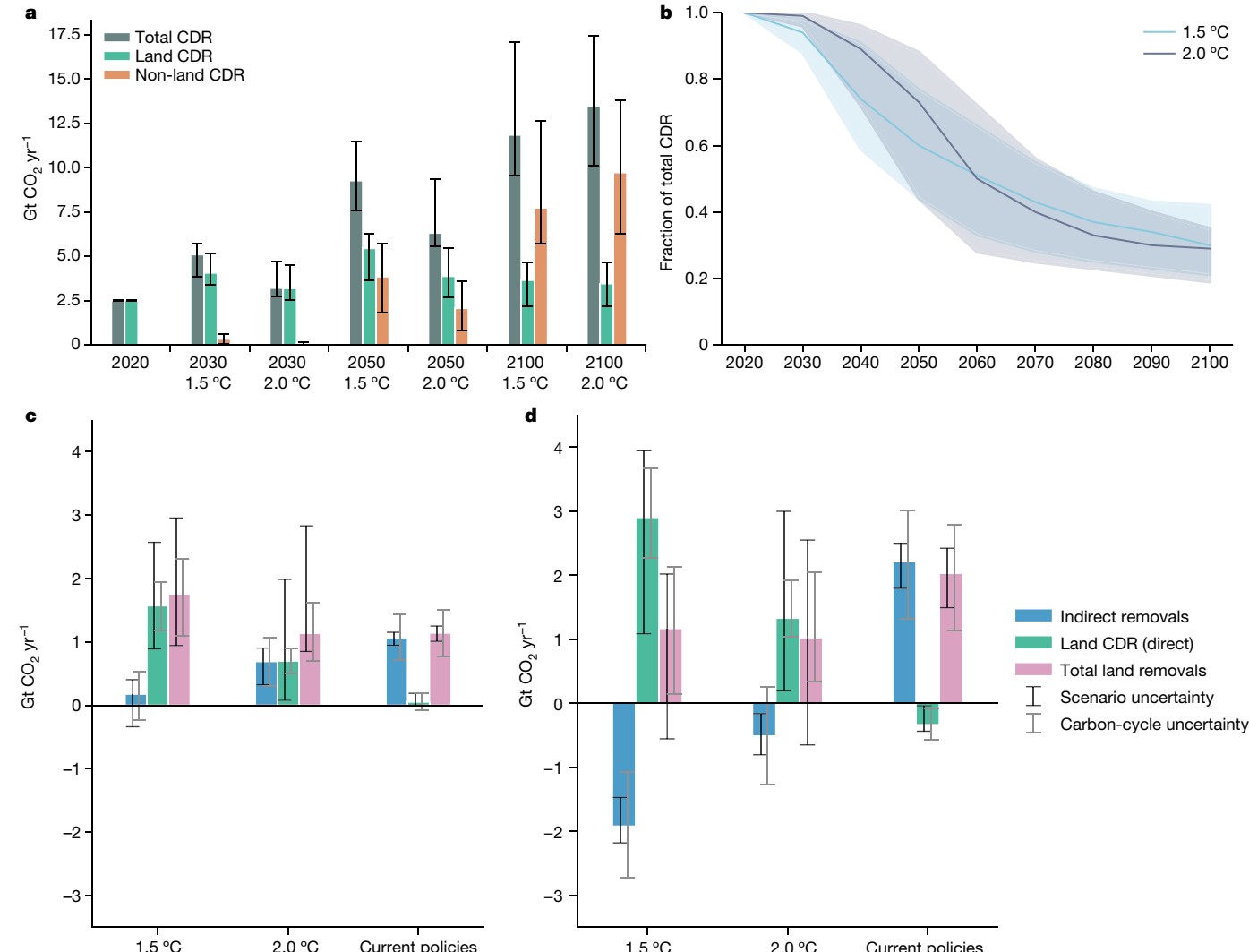

**Fig. 5 | CDR characteristics in mitigation and current-policy pathways.**
**a**, Net land-use carbon removal levels from direct fluxes (green bars) are
compared with non-land CDR (brown bars) and total levels (summing land-use
and CDR, grey bars) with whiskers denoting the interquartile range of each
estimate across 1.5 °C and 2.0 °C scenarios. Here, non-land CDR comprises
technologies included in the IAM pathways assessed in AR6 other than those
due solely to land-use change, such as bio-energy with carbon capture and
storage, direct air capture of $CO_2$ with storage and enhanced mineral
weathering. **b**, The share of land-based CDR reduces over time across both
1.5 °C and 2.0 °C pathways with the median (solid line) and interquartile range
(shaded area) shown for the population of scenarios assessed. The direct

component of land-based removal flux, which constitutes land-based CDR, and
the indirect component of the removal flux evolve differently across pathways.
**c**, In the near term, until 2030, the 1.5 °C pathways see a strong enhancement
of additional removals (pink bar), whereas the 2.0 °C pathways see a similar
addition of total removals as current-policy pathways. **d**, By mid-century,
additional removals in current-policy pathways out-pace both the 1.5 °C and
2.0 °C pathways, because of the continued enhancement of indirect removals
compared with an overall weakening of this flux in mitigation pathways. Scenario
uncertainty in **c**,**d** is estimated by the interquartile range of scenario-based
estimates, whereas the carbon-cycle uncertainty is estimated by the interquartile
range of the median ensemble of climate runs (Methods).

the UNFCCC to meaningfully compare aggregated national targets and
mitigation benchmarks. No matter what the reporting conventions
are, the near-term action that is needed to meet the Paris Agreement
is clear: emissions must peak as soon as possible and reduce markedly
this decade. This message must not be lost in the translation between
different concepts of anthropogenic land carbon fluxes.

## Online content

Any methods, additional references, Nature Portfolio reporting summa-
ries, source data, extended data, supplementary information, acknowl-
edgements, peer review information; details of author contributions
and competing interests; and statements of data and code availability
are available at https://doi.org/10.1038/s41586-023-06724-y.

1. den Elzen, M. G. J. et al. Updated nationally determined contributions collectively raise
   ambition levels but need strengthening further to keep Paris goals within reach. *Mitig.
   Adapt. Strateg. Glob. Change* **27**, 33 (2022).
2. Grassi, G. et al. Harmonising the land-use flux estimates of global models and national
   inventories for 2000–2020. *Earth Syst. Sci. Data* **15**, 1093–1114 (2023).
3. Friedlingstein, P. et al. Global Carbon Budget 2022. *Earth Syst. Sci. Data* **14**, 4811–4900
   (2022).
4. Grassi, G. et al. Critical adjustment of land mitigation pathways for assessing countries'
   climate progress. *Nat. Clim. Change* **11**, 425–434 (2021).
5. Gasser, T. et al. Historical $CO_2$ emissions from land use and land cover change and their
   uncertainty. *Biogeosciences* **17**, 4075–4101 (2020).
6. Ogle, S. M. & Kurz, W. A. Land-based emissions. *Nat. Clim. Change* **11**, 382–383 (2021).
7. UNFCCC. *Synthesis Report for the Technical Assessment Component of the First Global
   Stocktake* (UNFCCC, 2023).
8. van Beek, L., Oomen, J., Hajer, M., Pelzer, P. & van Vuuren, D. Navigating the political: an
   analysis of political calibration of integrated assessment modelling in light of the 1.5 °C
   goal. *Environ. Sci. Policy* **133**, 193–202 (2022).
9. UNFCCC. *Nationally Determined Contributions under the Paris Agreement. Revised Note
   by the Secretariat* (UNFCCC, 2021).

10. Höhne, N. et al. Wave of net zero emission targets opens window to meeting the Paris Agreement. *Nat. Clim. Change* **11**, 820–822 (2021).
11. Meinshausen, M. et al. Realization of Paris Agreement pledges may limit warming just below 2 °C. *Nature* **604**, 304–309 (2022).
12. Ou, Y. et al. Can updated climate pledges limit warming well below 2 °C? *Science* **374**, 693–695 (2021).
13. Rogelj, J. et al. Scenarios towards limiting global mean temperature increase below 1.5 °C. *Nat. Clim. Change* **8**, 325 (2018).
14. Roe, S. et al. Contribution of the land sector to a 1.5 °C world. *Nat. Clim. Change* **9**, 817–828 (2019).
15. Intergovernmental Panel on Climate Change. *Climate Change and Land: An IPCC Special Report on Climate Change, Desertification, Land Degradation, Sustainable Land Management, Food Security, and Greenhouse Gas Fluxes in Terrestrial Ecosystems* (Cambridge Univ. Press, 2022).
16. Grassi, G. et al. Reconciling global-model estimates and country reporting of anthropogenic forest $CO_2$ sinks. *Nat. Clim. Change* **8**, 914–920 (2018).
17. IPCC. in *Climate Change 2022: Mitigation of Climate Change. Working Group III Contribution to the Sixth Assessment Report of the Intergovernmental Panel on Climate Change* (eds Shukla, P. R. et al.) 3–48 (Cambridge Univ. Press, 2022).
18. UNFCCC. *Summary Report on the First Meeting of the Technical Dialogue of the First Global Stocktake under the Paris Agreement* (UNFCCC, 2022).
19. Houghton, R. A. & Nassikas, A. A. Global and regional fluxes of carbon from land use and land cover change 1850–2015. *Global Biogeochem. Cycles* **31**, 456–472 (2017).
20. Hansis, E., Davis, S. J. & Pongratz, J. Relevance of methodological choices for accounting of land use change carbon fluxes. *Global Biogeochem. Cycles* **29**, 1230–1246 (2015).
21. Buendia, E., Guendehou, S., Limmeechokchai, B. & Pipatti, R. *2019 Refinement to the 2006 IPCC Guidelines for National Greenhouse Gas Inventories* (UNFCCC, 2019).
22. Task Force on National Greenhouse Gas Inventories. *Revisiting the Use of Managed Land as a Proxy for Estimating National Anthropogenic Emissions and Removals* (UNFCCC, 2009).
23. Ogle, S. M. et al. Delineating managed land for reporting national greenhouse gas emissions and removals to the United Nations framework convention on climate change. *Carbon Balance Manage.* **13**, 9 (2018).
24. Canadell, J. G. et al. Factoring out natural and indirect human effects on terrestrial carbon sources and sinks. *Environ. Sci. Policy* **10**, 370–384 (2007).
25. IPCC. *2006 IPCC Guidelines for National Greenhouse Gas Inventories* (UNFCCC, 2008).
26. Grassi, G. et al. Carbon fluxes from land 2000–2020: bringing clarity to countries' reporting. *Earth Syst. Sci. Data* **14**, 4643–4666 (2022).
27. Mitchard, E. T. A. The tropical forest carbon cycle and climate change. *Nature* **559**, 527–534 (2018).
28. Lee, J.-Y. et al. in *Climate Change 2021: The Physical Science Basis. Contribution of Working Group I to the Sixth Assessment Report of the Intergovernmental Panel on Climate Change* (eds Masson-Delmotte, V. et al.) 553–672 (Cambridge Univ. Press, 2021).
29. Fyson, C. L. & Jeffery, M. L. Ambiguity in the land use component of mitigation contributions toward the Paris Agreement goals. *Earths Future* **7**, 873–891 (2019).
30. Forsell, N. et al. Assessing the INDCs' land use, land use change, and forest emission projections. *Carbon Balance Manag.* **11**, 26 (2016).
31. Schlamadinger, B. et al. A synopsis of land use, land-use change and forestry (LULUCF) under the Kyoto Protocol and Marrakech Accords. *Environ. Sci. Policy* **10**, 271–282 (2007).
32. Prütz, R., Strefler, J., Rogelj, J. & Fuss, S. Understanding the carbon dioxide removal range in 1.5 °C compatible and high overshoot pathways. *Environ. Res. Commun.* **5**, 041005 (2023).
33. Smith, S. M. et al. *The State of Carbon Dioxide Removal* 1st edn (The State of Carbon Dioxide Removal, 2023).
34. Byers, E. et al. AR6 scenarios database. *Zenodo* https://doi.org/10.5281/zenodo.5886912 (2022).
35. Jiang, M. et al. The fate of carbon in a mature forest under carbon dioxide enrichment. *Nature* **580**, 227–231 (2020).
36. Gasser, T., Ciais, P. & Lewis, S. L. How the Glasgow Declaration on Forests can help keep alive the 1.5 °C target. *Proc. Natl Acad. Sci. USA* **119**, e2200519119 (2022).
37. Gusti, M., Augustynczik, A. L. D., Di Fulvio, F., Lauri, P. & Forsell, N. in *The 2nd International Electronic Conference on Forests—Sustainable Forests: Ecology, Management, Products and Trade 23* (eds de Dios, V. R. & Dimopoulos, P.) (MDPI, 2021).
38. Fletcher, M.-S., Hamilton, R., Dressler, W. & Palmer, L. Indigenous knowledge and the shackles of wilderness. *Proc. Natl Acad. Sci. USA* **118**, e2022218118 (2021).
39. Ellis, E. C. et al. People have shaped most of terrestrial nature for at least 12,000 years. *Proc. Natl Acad. Sci. USA* **118**, e2023483118 (2021).

# Methods

## Selection of AR6 scenarios

As part of its Sixth Assessment Report, IPCC Working Group III authors analysed more than 2,200 scenarios for potential inclusion in its mitigation pathway assessment[40]. Of those, 1,202 were eventually vetted: deemed to have provided enough detail to allow a climate analysis using the climate assessment architecture of the IPCC[41]. Those scenarios were then divided into different scenario categories based on their peak and end-of-century temperature probabilities[34].

In this study, we focus on three scenarios: C1, C2 and C3 as defined in AR6 of the IPCC (ref. 40). C1 scenarios are as likely as not to limit warming to 1.5 °C and have been interpreted as consistent with the 1.5 °C long-term temperature goal of the Paris Agreement as outlined in Article 2 (ref. 42), although arguments have been made that further delineation should be made into scenarios that do and do not achieve net-zero $CO_2$ emissions to better reflect its Article 4 (ref. 43). We assess outcomes from the 2.0 °C C3 scenarios given their historic policy relevance, their capability to show progress towards 1.5 °C and their use in examining climate impacts beyond what is envisioned by the Paris Agreement. We also highlight mitigation outcomes of C2 scenarios, also called high overshoot scenarios, which are as likely as not to limit warming to 1.5 °C in 2100 but are likely to exceed 1.5 °C in the interim period. Such pathways are nominally similar in mitigation and impact assessment with C3 scenarios until at least mid-century[43].

For this analysis, we require that scenarios have been vetted by the IPCC climate analysis framework and provide a minimum set of land-cover variables such as Land Cover|Cropland, Land Cover|Forestry and Land Cover|Pasture. We analyse the presence of each of these variables and their combination in Extended Data Table 3 at the global, IPCC 5-region (R5) and IPCC 10-region (R10) levels. Balancing concerns of greater regional detail and greater scenario coverage, we perform our analysis based on the R5 regions (Extended Data Table 4) given that nearly all models with full global variable coverage also provide detail at the R5 regional level for the C1–C3 scenarios.

To understand how well our scenario subset containing R5 land-cover variables corresponds statistically to the full database sample of the C1–C3 scenarios, we perform a Kolmogorov–Smirnov test over key mitigation variables of interest including GHG and $CO_2$ 2030 emission reductions, median peak warming, median warming in 2100, year of median warming, cumulative net $CO_2$ emissions throughout the century, cumulative net $CO_2$ until net-zero and cumulative net negative $CO_2$ after net-zero (Extended Data Fig. 7). For all variables, the Kolmogorov–Smirnov test is not able to determine whether the R5 subset comes from a different distribution than the full database sample, whereas it is able to determine the non-R5 subset is different for peak warming and cumulative net $CO_2$ emissions, both of which are shown in Extended Data Fig. 8. These results indicate that the subset of about 75–80% of all the C1–C3 scenarios we chose to perform our analysis will result in sufficiently similar macro-mitigation outcomes to represent such outcomes from the original distribution of scenarios.

## Reanalysis with OSCAR

We use OSCAR v.3.2: a version structurally similar to the one used for the 2021 Global Carbon Budget (GCB)[44], albeit used here with a regional aggregation that matches the R5 IPCC regions. We first run a historical simulation (starting in 1750 and ending in 2020) using the same experimental setup as for the 2021 GCB[5,44], with the updated input data used in ref. 36. This historical simulation is used not only to initialize the model in 2014 for the scenario simulations but also to constrain the Monte Carlo ensemble ($n$ = 1,200) using two values (instead of one in the GCB): the cumulative land carbon sink in the absence of land-cover change over 1960–2020 and the NGHGI-compatible emissions averaged over

2000–2020. The former is a constraint of $135 \pm 25$ Gt $CO_2$ yr$^{-1}$ (ref. 44). The latter is a constraint of $-0.45 \pm 0.77$ Gt $CO_2$ yr$^{-1}$, using ref. 2 as a central estimate and combining uncertainties in ELUC and SLAND from the GCB. All physical uncertainties in this section are 1 standard deviation ($1\sigma$). All values reported in the main text and figures are obtained using the weighted average and standard deviation of the Monte Carlo ensemble, using these two constraints for the weighting[5].

To run the final scenario simulations over 2014–2100, OSCAR needs two types of input data: (1) $CO_2$ and local climate projections and (2) land use and land-cover change projections. The former mostly affects the land carbon sink (that is, the indirect effect), whereas the latter mostly affects the bookkeeping emissions (that is, the direct effect). OSCAR follows a theoretical framework[45] that enables a clear separation of both direct and indirect effects. Only the direct effect is reported annually in the GCB. Note that we do not re-evaluate the land-cover change albedo effect because this was already included in the original AR6 database climate projections.

Atmospheric $CO_2$ time series is taken directly from the database, as the median outcome estimated by the Model for the Assessment of Greenhouse Gas Induced Climate Change (MAGICC). However, local climate temperature and precipitation changes are not directly available. These are, therefore, computed using the internal equations of OSCAR[46], and the time series of global temperature change and species-based effective radiative forcing (ERF) from the database (same source). The missing components of the global ERF were treated as follows. Black carbon on snow and stratospheric $H_2O$ start at a historical level in 2014 (ref. 47) and follow the same relative annual change as the ERF of the scenario from black carbon and $CH_4$, respectively. Contrails are assumed constant after 2014. Solar forcing is assumed to follow the same pathway common to all Shared Socioeconomic Pathways (SSPs). Volcanic aerosols are assumed to be constant and equal to the average of the historical period (that is, to have a zero ERF). Finally, we apply a linear transition over 2014–2020 between the observed and projected $CO_2$ and climate, so that these variables are 100% observed in 2014 and 100% projected in 2020. We note that the observed and projected $CO_2$ are virtually indistinguishable over that period but the observed and projected regional climate changes do differ by up to a few tenths of a degree. We further note that, because only median atmospheric $CO_2$, ERF and global temperature are used as input, we do not sample and report the full physical uncertainty of the Earth system, but only the biogeochemical uncertainty from the terrestrial carbon cycle in response to these median outcomes.

Land use and land-cover change input data for OSCAR have three variables: the land cover change per se, wood harvest data (expressed in carbon amount taken from woody areas without changing the land cover) and shifting cultivation (a traditional activity consisting of cycles of cutting forest for agriculture, abandoning to recover soil fertility and then returning). Wood harvest and shifting cultivation information are not provided in the database; so we use proxy variables to extrapolate the historical 2014 values. Wood harvest is scaled using the Forestry Production|Roundwood variable, and shifting cultivation is scaled using Primary Energy|Biomass|Traditional as a proxy of the development level of a region. When scenarios did not report these proxy variables, we assumed a constant wood harvest or shifting cultivation in the future, because these are second-order effects on the global bookkeeping emissions.

Land-cover change is split between gains and losses that are deduced directly as the year-to-year difference (gain if positive, loss if negative) using the following land-cover variables of the database: Land Cover|Forest, Land Cover|Cropland, Land Cover|Pasture and Land Cover|Built-up Area (built-up area is assumed to be constant if not available). Land-cover change in the remaining biome of OSCAR (non-forested natural land) is deduced afterwards to maintain a constant land area. To build the transitions matrix required as input by OSCAR, it is then assumed that the area increase of a given biome

occurs at the expense of all the biomes that see an area decrease (within the same region and at the same time step), in proportion to the share of total area decrease of the biomes. By construction, this approach provides only net land-cover transitions because it is impossible to have gain and loss in the same year, in a given region. Therefore, and because our historical data account for gross transitions but scenarios do not, we add to this net transitions matrix a constant amount of reciprocal transitions equal to their average historical value over 2008–2020 to obtain a gross transitions matrix. Finally, the three land use and land-cover change input variables follow the same linear transition over 2014–2020 as the $CO_2$ and climate forcings.

We extract two key variables (and their subcomponents) from these scenario simulations: the bookkeeping emissions (ELUC in the GCB) and the land carbon sink (SLAND in the GCB). Following the approach in ref. 4, the adjustment flux (that is, the indirect flux included in the NGHGIs but not included by the IAMs, also called the factor in the main text) required to move from bookkeeping emissions to NGHGI-compatible emissions is calculated as the part of the land carbon sink that occurs in forests that are managed. Therefore, we obtain the adjustment flux by multiplying the value of SLAND simulated for forests by the fraction of (officially) managed forests. We set this fraction to the one estimated by ref. 4 for 2015, which also allows us to deduce the area of managed and unmanaged (that is, intact) forest in our base year. We then estimate how the area of intact forest evolves in each scenario, assuming that forest gains are always managed forest (that is, they do not change intact forest area) and that half of the forest losses are losses of intact forest with the other half being losses of the managed forest. This fraction is deduced from ref. 48 that estimated that around 92 Mha of intact forest disappeared between 2000 and 2013, whereas the FAO Global Forest Resources Assessment 2020 reports about 170 Mha of gross deforestation over the same period. We acknowledge, however, that applying a global and constant value for this fraction is a coarse approximation that should be refined in future work, possibly using information from the scenario database itself. This assumption also implies that, as long as there is a background gross deforestation (as is the case here, given the added reciprocal transitions), countries will report more and more managed forest area. This is not necessarily inconsistent with the Glasgow Declaration on Forest made at COP26, as its implications in terms of pristine forest conservation are unclear[36]. The subcomponents of the bookkeeping emissions are extracted following the land categories defined in ref. 2, and we consider that the net flux happening in the forest land category, excluding shifting cultivation, is the direct contribution to land CDR. The indirect contribution to land CDR would be exactly the adjustment flux described above.

The re-analysed bookkeeping net emissions (that is, direct effect) show an average deviation of −87 Gt $CO_2$ for C1 scenarios and −63 Gt $CO_2$ for C3 scenarios from the reported emissions in the database, accumulated over the course of the century. Using the best-guess transient-climate response to cumulative emissions estimated by the IPCC (ref. 49), this implies that the global temperature outcomes of these scenarios would differ by about −0.04 °C and −0.03 °C, respectively, from what was reported in the IPCC report, if our estimates of bookkeeping emissions were used instead of those reported by the IAM teams.

Furthermore, after re-allocating the indirect effect in managed forest (to align with the NGHGIs), we observe a $4.4 \pm 1.0$ Gt $CO_2$ yr$^{-1}$ difference between the aligned and unaligned historical LULUCF emissions over 2000–2020. This number is at the lower end of the latest $6.4 \pm 1.2$ Gt $CO_2$ yr$^{-1}$ provided in the 2022 GCB[3]. Compared with the $6.7 \pm 2.5$ Gt $CO_2$ yr$^{-1}$ difference reported in ref. 2, and correcting for the absence of organic soils emissions in our simulations with OSCAR (about 0.8 Gt $CO_2$ yr$^{-1}$), OSCAR can explain about 75% of the observed difference. Although OSCAR typically produces fairly central estimates

of the direct effect[3], its estimates of the indirect effect show a biased high $CO_2$ fertilization[50].

## Comparing adjusted pathways with current policy and NDC estimates

We use the latest available estimate of aggregate NDCs from ref. 1 to compare with the NGHGI-adjusted global pathways. The 1.5 °C and 2.0 °C pathways we use are the same as previously discussed: the IPCC C1 and C3 pathways with sufficient land cover detail at the R5 region. We additionally re-analyse the current-policy pathways from the IPCC AR6 database. These correspond to pathways consistent with the current policies as assessed by the IPCC, or the P1b pathways as per the AR6 database metadata indicator Policy_category_name.

We incorporate an endogenous estimation of the indirect effect with OSCAR, which varies over time based on land-cover pattern changes and changes to carbon-cycle dynamics and carbon fertilization. As such, we compare our central estimate of global GHG emissions in 2015, approximately 49.4 Gt $CO_2$-equiv to that in ref. 1, 51.2 Gt $CO_2$-equiv, resulting in a difference of 1.8 Gt $CO_2$-equiv. We then apply this offset value (1.8 Gt) to all estimations of 2030 emission levels in ref. 1 to provide comparable levels with our pathways. This ensures that the NDC targets calculated based on national inventories become comparable with the NGHGI-adjusted modelled pathways.

## Data availability

All data generated and analysed here are available at GENIE Scenario Explorer (https://data.ece.iiasa.ac.at/genie).

## Code availability

OSCAR is an open-source model available at GitHub (https://github.com/tgasser/OSCAR). Source code for all analysis files is available at GitHub (https://github.com/iiasa/gidden_ar6_reanalysis).

40. Riahi, K. et al. in *IPCC, 2022: Climate Change 2022: Mitigation of Climate Change. Contribution of Working Group III to the Sixth Assessment Report of the Intergovernmental Panel on Climate Change* (eds Shukla, P. R. et al.) (Cambridge Univ. Press, 2022).
41. Kikstra, J. S. et al. The IPCC Sixth Assessment Report WGIII climate assessment of mitigation pathways: from emissions to global temperatures. *Geosci. Model Dev.* **15**, 9075–9109 (2022).
42. Mace, M. J. Mitigation commitments under the Paris Agreement and the way forward. *Climate Law* **6**, 21–39 (2016).
43. Schleussner, C.-F., Ganti, G., Rogelj, J. & Gidden, M. J. An emission pathway classification reflecting the Paris Agreement climate objectives. *Commun. Earth Environ.* **3**, 135 (2022).
44. Friedlingstein, P. et al. Global Carbon Budget 2021. *Earth Syst. Sci. Data* **14**, 1917–2005 (2022).
45. Gasser, T. & Ciais, P. A theoretical framework for the net land-to-atmosphere $CO_2$ flux and its implications in the definition of "emissions from land-use change". *Earth Syst. Dyn.* **4**, 171–186 (2013).
46. Gasser, T. et al. The compact Earth system model OSCAR v2.2: description and first results. *Geosci. Model Dev.* **10**, 271–319 (2017).
47. Smith, C. et al. in *Climate Change 2021: The Physical Science Basis. Contribution of Working Group I to the Sixth Assessment Report of the Intergovernmental Panel on Climate Change* (eds Masson-Delmotte, V. et al.) 923–1054 (Cambridge Univ. Press, 2021).
48. Potapov, P. et al. The last frontiers of wilderness: Tracking loss of intact forest landscapes from 2000 to 2013. *Sci. Adv.* **3**, e1600821 (2017).
49. Canadell, J. G. et al. in *Climate Change 2021: The Physical Science Basis. Contribution of Working Group I to the Sixth Assessment Report of the Intergovernmental Panel on Climate Change* (eds Masson-Delmotte, V. et al.) 673–816 (Cambridge Univ. Press, 2021).
50. Quilcaille, Y., Gasser, T., Ciais, P. & Boucher, O. CMIP6 simulations with the compact Earth system model OSCAR v3.1. *Geosci. Model Dev.* **16**, 1129–1161 (2023).

**Acknowledgements** We thank M. Sanz and C.-F. Schleussner for the comments on an initial draft of the paper, M. Beer and F. Spagopoulou from the Designers for Climate for expert support on visualizations, and the reviewers whose comments improved the quality of the paper. We acknowledge the funds received from the Horizon Europe research of the European Union and the innovation programme RESCUE, grant agreement no. 101056939 (M.J.G. and T.G.); the Horizon 2020 research of the European Union and the innovation programme ESM2025— Earth System Models for the Future, grant no. 101003536 (T.G. and Z.N.); and the ERC-2020-SyG

GENIE grant of the European Union, grant no. 951542 (M.J.G., W.F.L., J.M. and K.R.). The views expressed are those of the writers and may not under any circumstances be regarded as stating an official position of the European Commission.

**Author contributions** M.J.G., T.G. and K.R. contributed to the conceptualization; M.J.G., T.G., G.G., I.J. and Z.N. devised the methodology; M.J.G. and T.G. helped in the investigation; T.G. provided the software support; M.J.G. helped with visualization; M.J.G. wrote the original draft; and M.J.G., T.G., G.G., N.F., I.J., W.F.L., J.M., Z.N., J.S. and K.R. reviewed and edited the paper.

**Competing interests** The authors declare no competing interests.

**Additional information**
**Correspondence and requests for materials** should be addressed to Matthew J. Gidden.

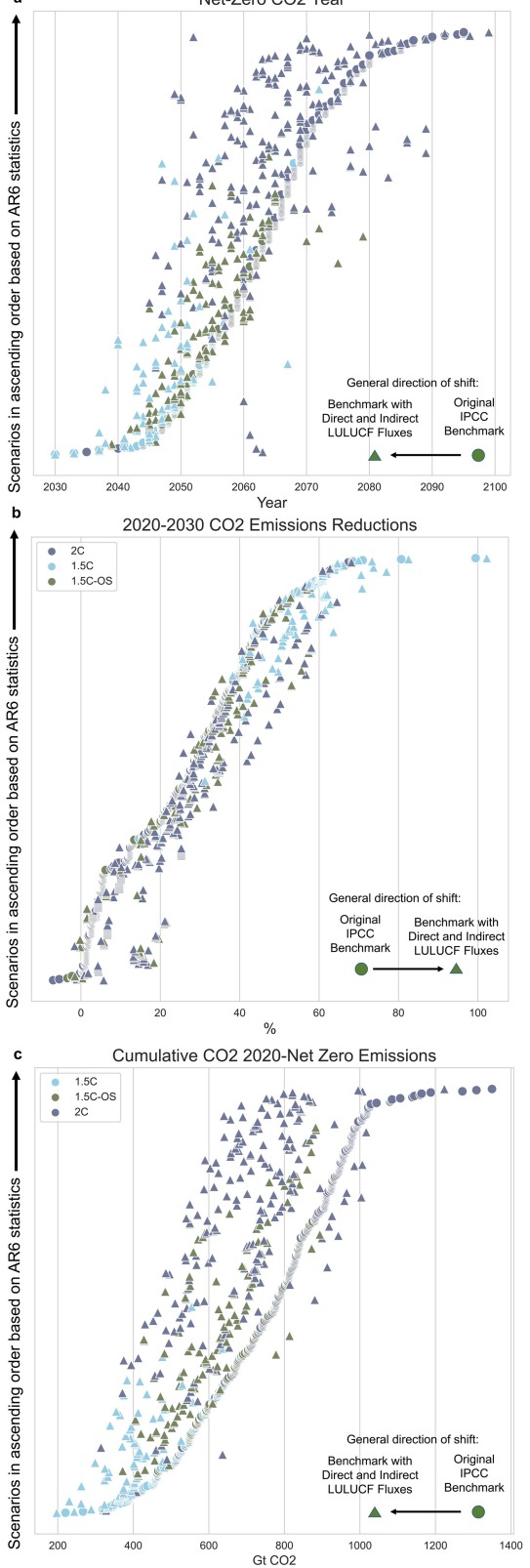

**Extended Data Fig. 1 | Scenario-wise mitigation benchmark shift.** The change between estimates of mitigation benchmarks for 1.5 C (blue, IPCC category C1), 1.5C-OS (green, IPCC category C2), and 2 C (purple, IPCC category C3) scenarios. Original values from the AR6 database (which follows IAM reporting conventions) are shown as circles whereas values derived from reanalyzed scenarios in this study (in line with NGHGI reporting conventions) are shown as triangles. The estimates of the year of global net-zero $CO_2$ (panel a), emissions reductions between 2020 and 2030 (panel b), and cumulative $CO_2$ emissions (panel c) are shown. Each pair of dots and triangles represents results from a single scenario, with scenarios ordered along the y-axis based on the values in the original AR6 dataset.

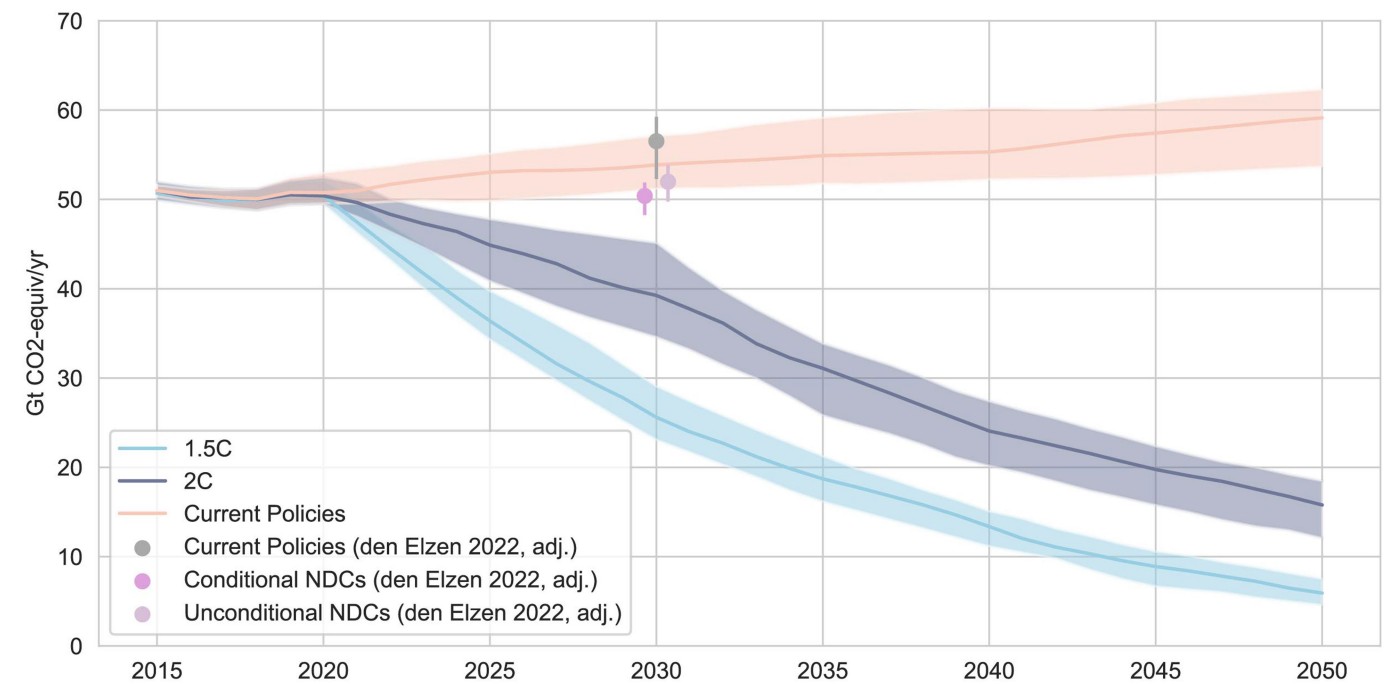

**Extended Data Fig. 2 | NGHGI-adjusted global GHG pathways compared with NDCs and current policies.** The interquartile range shown and median highlighted is plotted together with current estimates of 2030 aggregated national climate target levels and current policy estimates from den Elzen et al. (2022).

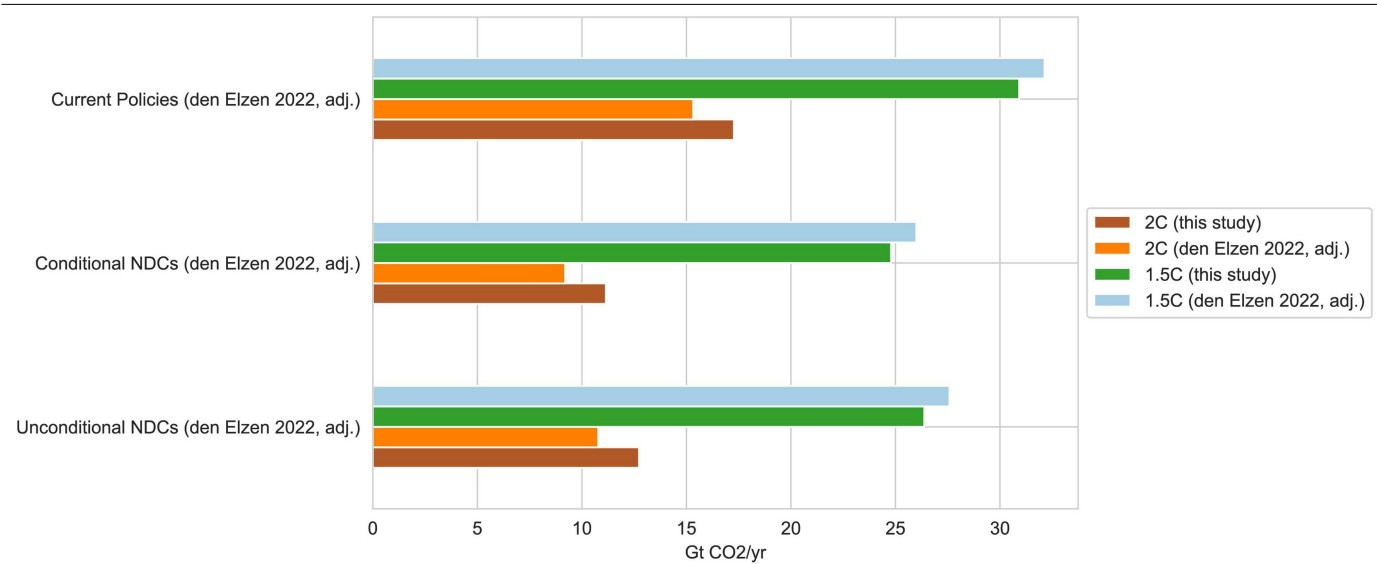

**Extended Data Fig. 3 | The 2030 emissions gap between current policies and pledges.** 1.5 C and 2 C as assessed in this study and by den Elzen (2022) is compared against levels of current policies, conditional NDCs, and unconditional NDCs as reported in den Elzen (2022). Median estimates of all values are used to compute the respective emission gaps.

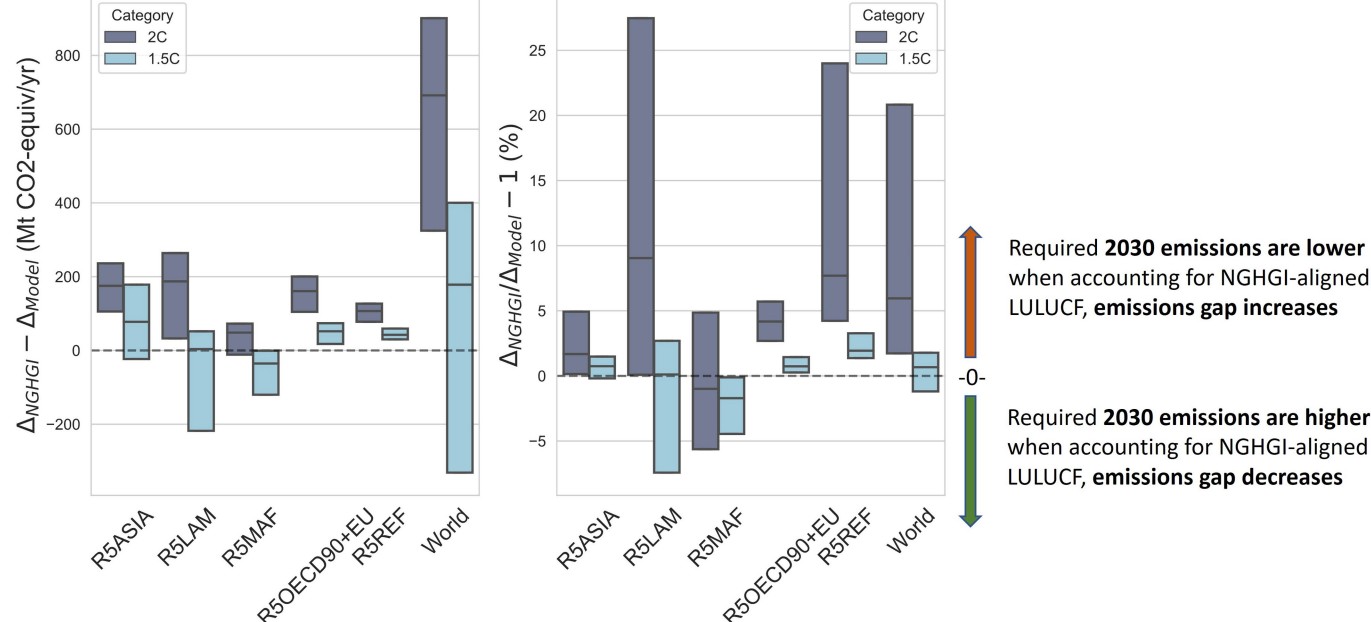

**Extended Data Fig. 4 | The change in the estimated 2030 emission gap between due to alignment to NGHGI conventions.** The total magnitude, left, relative value, right. Each bar represents the median value with the interquartile range of the estimate across scenarios. These changes occur differently across different regions between pathways following model-based conventions and adjusted pathways following NGHGI-based conventions. A positive value means that the gap is larger when considering both (i.e. when aligned to NGHGIs), and a negative value means the gap is smaller. Regions labels conform to IPCC 5-region labels for Asia, Latin America, Middle East and Africa, the OECD and EU, and the Reformed Economies, respectively (Extended Data Table 4).

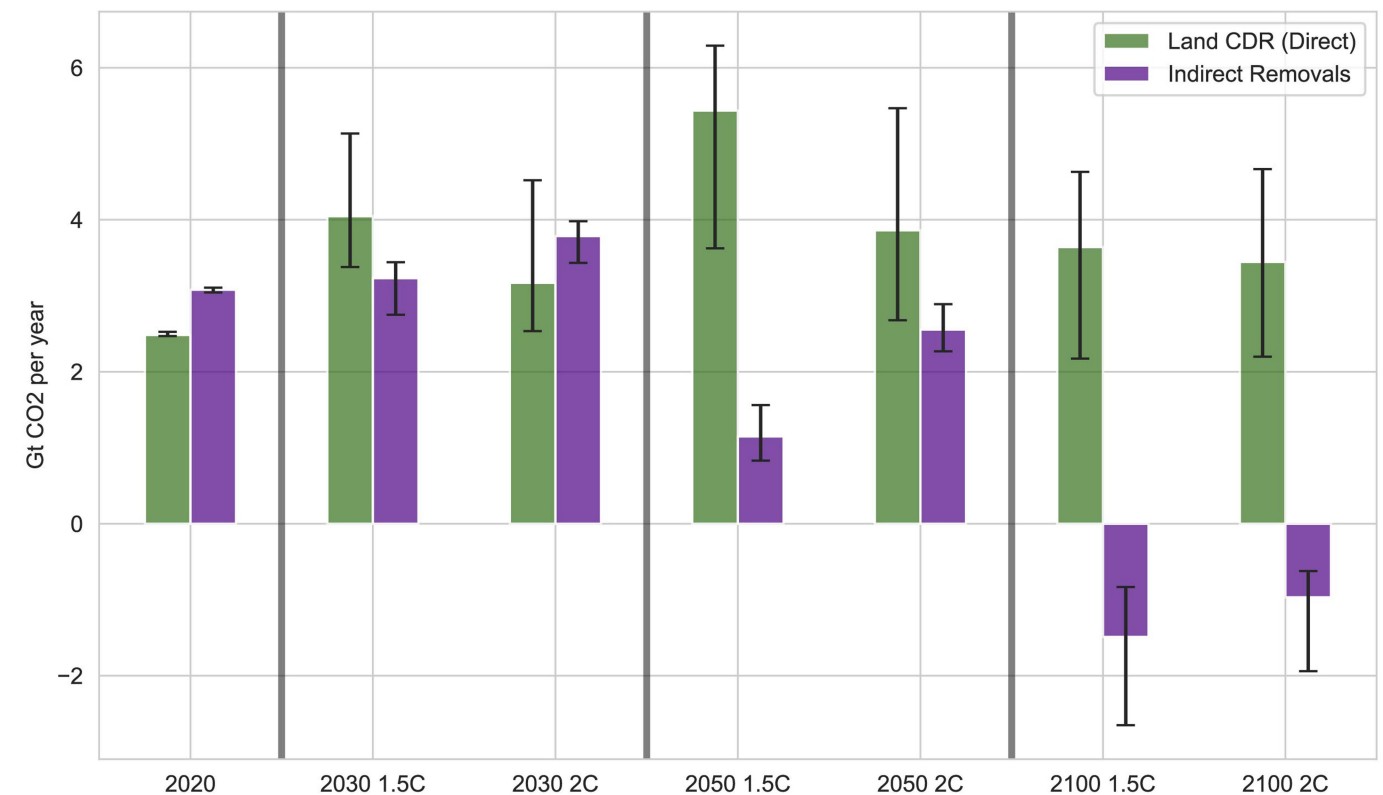

**Extended Data Fig. 5 | Gross carbon removal levels.** Gross carbon removal levels from LULUCF (reanalyzed with OSCAR) by direct effects (green) and indirect effects (purple) across 1.5 C and 2 C pathways. Interquartile ranges of each estimate are shown by error bars.

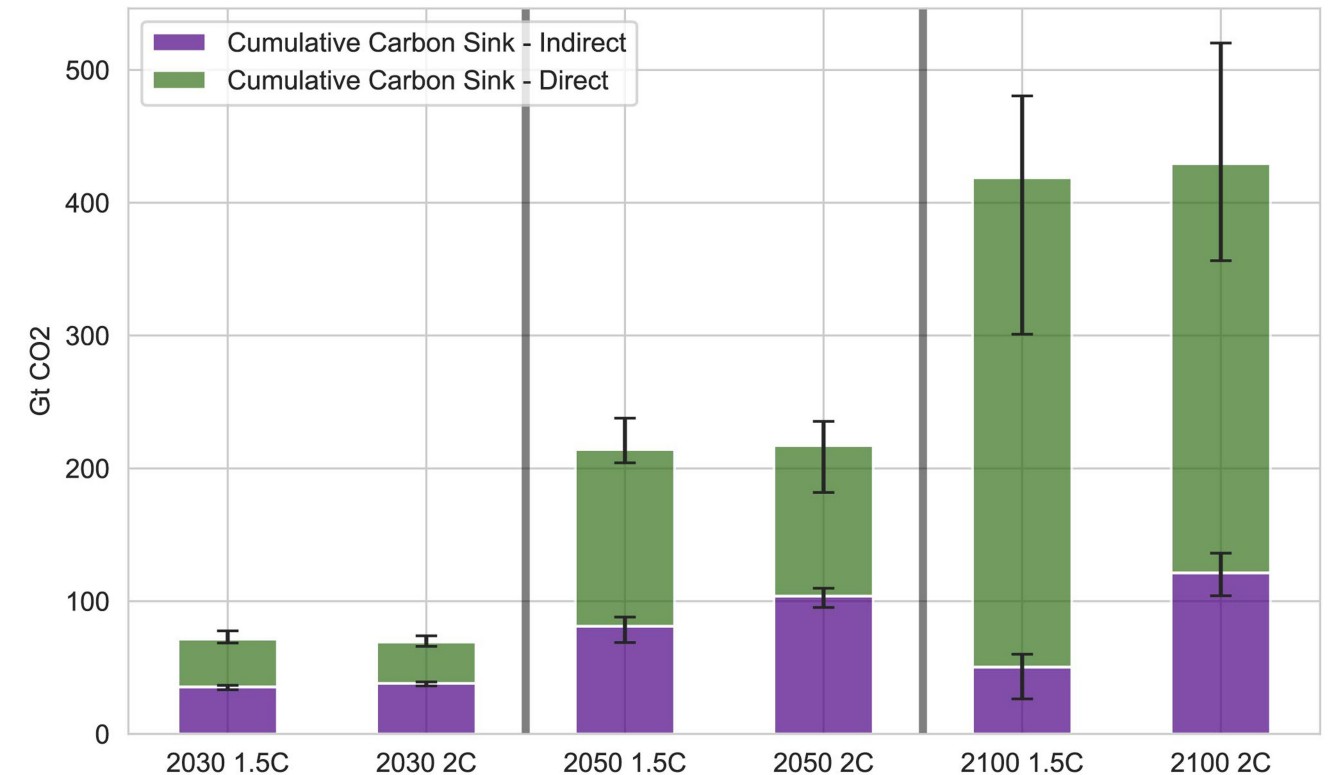

**Extended Data Fig. 6 | Cumulative carbon sequestered on land starting from 2020.** Gross cumulative carbon removal levels starting from 2020 from LULUCF (reanalyzed with OSCAR) by direct effects (green) and indirect effects (purple) across 1.5 C and 2 C pathways. Removals in both categories increase by midcentury, but at different levels. Both pathway categories see similar total cumulative removal levels by the end of the century with varying strength of indirect removals.

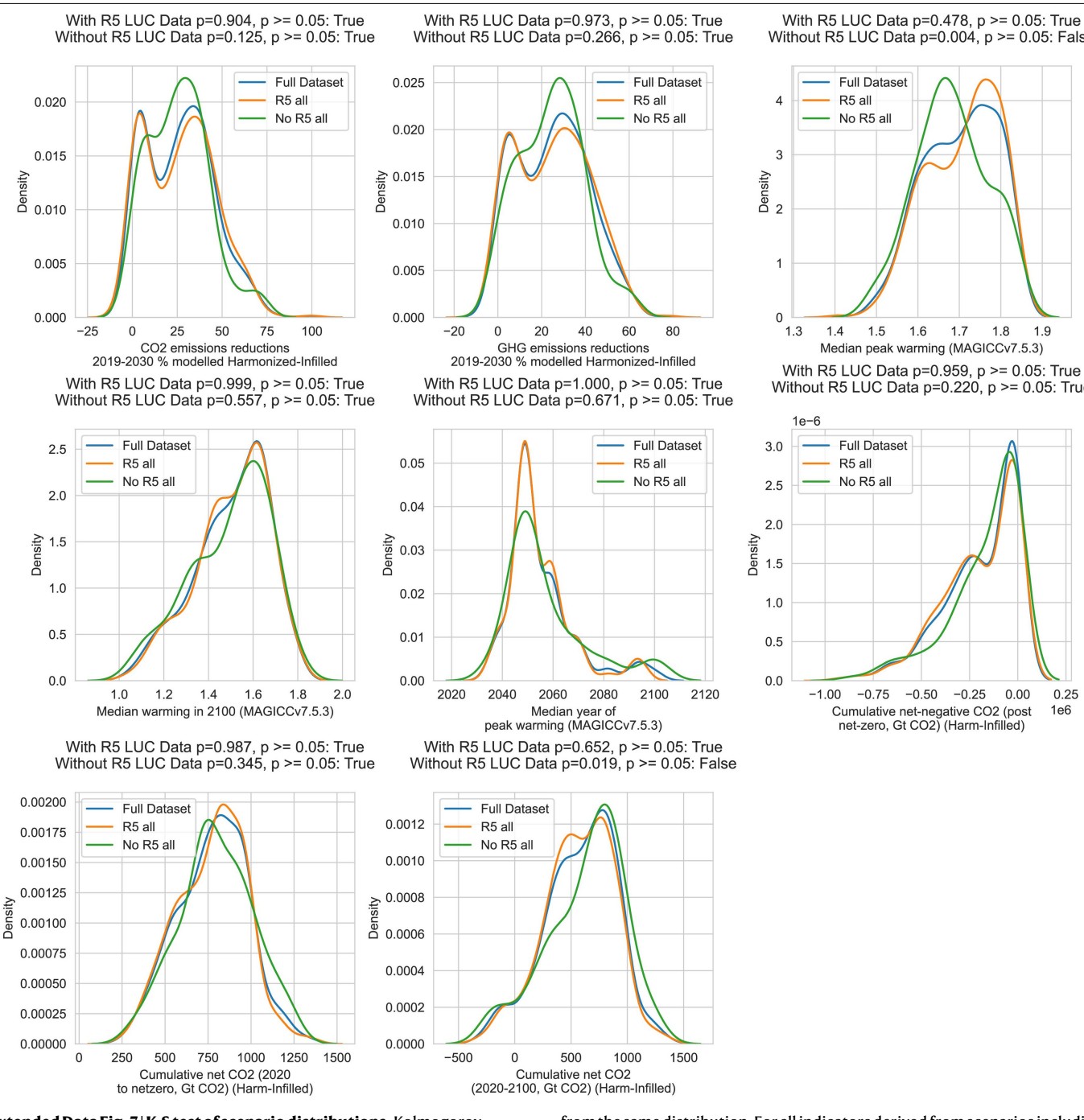

**Extended Data Fig. 7 | K-S test of scenario distributions.** Kolmogorov-Smirnov (K-S) test results for key mitigation indicators for the full set of C1-C3 scenarios, those scenarios having all land-cover variables defined at the R5 region level, and those not having all land-cover variables defined at the R5 region level. The null hypothesis of the K-S test is that two dataset values are from the same distribution. For all indicators derived from scenarios including land-cover variables data at the R5 region level, we can not reject the null hypothesis (p > 0.05). Some indicators of the scenario set without land-cover data (not used in this analysis) do reject the null hypothesis.

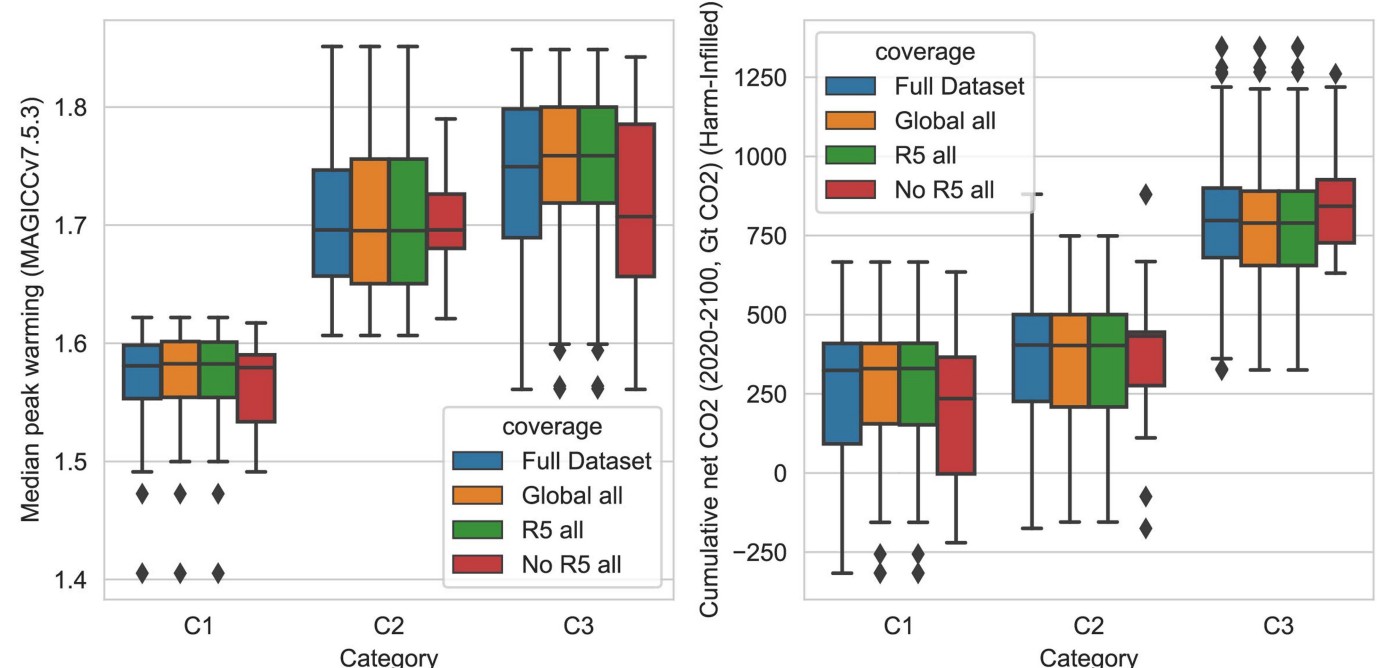

**Extended Data Fig. 8 | Mitigation metrics from scenario subsets.** Key mitigation metrics where scenarios without R5 region coverage (in red) cannot replicate the full database outcome. The blue bar presents the outcome for the full database, scenarios with global values of land-cover variables and R5 values are shown in yellow, and scenarios with land-cover variables at the R5 region are shown in green. The red bar shows how the distribution changes when considering the population of scenarios without full variable coverage ('No R5 all').

**Extended Data Table 1 | Indirect LULUCF flux estimates aligned with NGHGIs**

| IPCC Category | Year | R5ASIA | R5LAM | R5MAF | R5OECD90+EU | R5REF |
|---|---|---|---|---|---|---|
| 1.5°C, C1 | 2025 | -1.03 [-1.08, -0.99] | 0.27 [0.22, 0.37] | -0.49 [-0.52, -0.46] | -1.12 [-1.15, -1.11] | -0.85 [-0.86, -0.85] |
| | 2050 | -0.39 [-0.50, -0.28] | 0.48 [0.38, 0.60] | -0.07 [-0.12, -0.02] | -0.64 [-0.69, -0.57] | -0.58 [-0.61, -0.54] |
| | 2075 | 0.23 [0.16, 0.40] | 0.50 [0.39, 0.68] | 0.19 [0.15, 0.31] | -0.03 [-0.09, 0.10] | -0.20 [-0.23, -0.12] |
| | 2100 | 0.46 [0.25, 0.71] | 0.48 [0.33, 0.79] | 0.24 [0.17, 0.41] | 0.25 [0.13, 0.48] | 0.02 [-0.05, 0.13] |
| 2°C, C3 | 2025 | -1.05 [-1.09, -1.03] | 0.16 [0.12, 0.26] | -0.53 [-0.54, -0.51] | -1.15 [-1.17, -1.14] | -0.87 [-0.88, -0.86] |
| | 2050 | -0.77 [-0.89, -0.68] | 0.24 [0.15, 0.33] | -0.27 [-0.31, -0.22] | -0.98 [-1.05, -0.91] | -0.79 [-0.83, -0.74] |
| | 2075 | -0.04 [-0.12, 0.04] | 0.38 [0.30, 0.46] | 0.08 [0.03, 0.12] | -0.33 [-0.40, -0.24] | -0.40 [-0.44, -0.35] |
| | 2100 | 0.31 [0.23, 0.56] | 0.50 [0.39, 0.71] | 0.21 [0.14, 0.39] | 0.09 [0.03, 0.30] | -0.11 [-0.15, -0.02] |

Median values and interquartile ranges of the indirect flux in Gt $CO_2 yr^{-1}$ estimated by OSCAR per R5 IPCC region (see Extended Data Table 4 for region definitions). This value is computed for every scenario with sufficient land-use data (see Methods) in each model region for every point in time. This value constitutes the 'NGHGI Adjustment Factor' and is computed and added to each scenarios' estimated direct LULUCF flux values to quantify emissions pathways from global models aligned with NGHGI LULUCF reporting conventions.

**Extended Data Table 2 | Updated mitigation benchmarks**

| | 1.5°C, C1 | | | 2°C, C3 | | |
|---|---|---|---|---|---|---|
| | **(a)** | **(b)** | **(c)** | **(a)** | **(b)** | **(c)** |
| Cumulative net $CO_2$ emissions from 2020 until net-zero $CO_2$ (Gt CO2) | 512 (328-708) | 473 (319-620) | 392 (262-528) | 882 (635-1133) | 838 (542-1100) | 703 (445-936) |
| CO2 Emissions Reductions (2020-2030) (Gt CO2 yr$^{-1}$) | 47 (36-69) | 52 (35-67) | 56 (39-73) | 21 (1-43) | 21 (1-50) | 25 (4-55) |
| Net-zero CO2 Year | 2052 (2037-2067) | 2050 (2040-2060) | 2047 (2037-2056) | 2070 (2059-2093) | 2068 (2052-2087) | 2064 (2049-2083) |
| Net-zero GHG Year | 2098 (2054-2100) | 2066 (2051-2087) | 2067 (2049-2087) | 2100 (2078-2100) | 2082 (2069-2096) | 2082 (2066-2097) |

Net mitigation outcomes from scenarios: (**a**) as assessed by the IPCC in AR6, (**b**) with direct effects of LULUCF reanalyzed by OSCAR, and (**c**) including both direct and indirect effects of LULUCF (i.e. aligned to NGHGIs). All values provided as medians with 5$^{th}$–95$^{th}$ percentile ranges in parentheses.

**Extended Data Table 3 | Variable coverage of scenarios**

| Category | C1 | C2 | C3 | C4 | C5 | C6 | C7 | C8 |
|---|---|---|---|---|---|---|---|---|
| **Global Land Cover\|Forest** | **77%** | 80% | **77%** | 88% | 89% | 84% | 61% | 31% |
| **Global Land Cover\|Pasture** | **74%** | 80% | **75%** | 87% | 88% | 84% | 60% | 31% |
| **Global Land Cover\|Cropland** | **74%** | 80% | **75%** | 87% | 88% | 84% | 60% | 31% |
| **Global all** | **74%** | 80% | **75%** | 87% | 88% | 84% | 60% | 31% |
| **R5 Land Cover\|Forest** | **76%** | 80% | **77%** | 88% | 89% | 84% | 60% | 31% |
| **R5 Land Cover\|Pasture** | **73%** | 80% | **75%** | 87% | 88% | 84% | 60% | 31% |
| **R5 Land Cover\|Cropland** | **73%** | 80% | **75%** | 87% | 88% | 84% | 60% | 31% |
| **R5 all** | **73%** | 80% | **75%** | 87% | 88% | 84% | 60% | 31% |
| **R10 Land Cover\|Forest** | **59%** | 63% | **56%** | 57% | 66% | 56% | 30% | 17% |
| **R10 Land Cover\|Pasture** | **59%** | 62% | **56%** | 57% | 66% | 56% | 30% | 17% |
| **R10 Land Cover\|Cropland** | **59%** | 63% | **56%** | 57% | 66% | 56% | 30% | 17% |
| **R10 all** | **59%** | 62% | **56%** | 57% | 66% | 56% | 30% | 17% |

Fraction of AR6 database scenarios with land-use variables of interest, per scenario category.

**Extended Data Table 4 | Regional definitions**

| Macro Region | Short Name | Country Constituents |
|---|---|---|
| R5ASIA | Asia | China, China Hong Kong SAR, China Macao SAR, Mongolia, Taiwan, Afghanistan, Bangladesh, Bhutan, India, Maldives, Nepal, Pakistan, Sri Lanka, Brunei Darussalam, Cambodia, Democratic People's Republic of Korea, East Timor, Indonesia, Lao People's Democratic Republic, Malaysia, Myanmar, Papua New Guinea, Philippines, Republic of Korea, Singapore, Thailand, Viet Nam |
| R5LAM | Latin American | Argentina, Bahamas, Barbados, Belize, Bolivia, Brazil, Chile, Colombia, Costa Rica, Cuba, Dominican Republic, Ecuador, El Salvador, Guadeloupe, Guatemala, Guyana, Haiti, Honduras, Jamaica, Martinique, Mexico, Netherlands Antilles, Nicaragua, Panama, Paraguay, Peru, Puerto Rico, Suriname, Trinidad and Tobago, Uruguay, Venezuela |
| R5MAF | Middle East and Africa | Bahrain, Iran (Islamic Republic of), Iraq, Israel, Jordan, Kuwait, Lebanon, Oman, Qatar, Saudi Arabia, Syrian Arab Republic, United Arab Emirates, Yemen, Algeria, Angola, Benin, Botswana, Burkina Faso, Burundi, Cote d'Ivoire, Cameroon, Cape Verde, Central African Republic, Chad, Comoros, Congo, Democratic Republic of the Congo, Djibouti, Egypt, Equatorial Guinea, Eritrea, Ethiopia, Gabon, Gambia, Ghana, Guinea, Guinea-Bissau, Kenya, Lesotho, Liberia, Libyan Arab Jamahiriya, Madagascar, Malawi, Mali, Mauritania, Mauritius, Morocco, Mozambique, Namibia, Niger, Nigeria, Reunion, Rwanda, Senegal, Sierra Leone, Somalia, South Africa, Sudan, Swaziland, Togo, Tunisia, Uganda, United Republic of Tanzania, Western Sahara, Zambia, Zimbabwe |
| R5OECD90+EU | OECD90 and EU (and EU candidate) | Albania, Austria, Belgium, Bosnia and Herzegovina, Bulgaria, Croatia, Cyprus, Czech Republic, Denmark, Estonia, Finland, France, Germany, Greece, Hungary, Iceland, Ireland, Italy, Latvia, Lithuania, Luxembourg, Macedonia, Malta, Montenegro, Netherlands, Norway, Poland, Portugal, Spain, Sweden, Switzerland, Turkey, United Kingdom, Canada, United States of America, Australia, Fiji, French Polynesia, Guam, Japan, New Caledonia, New Zealand, Romania, Samoa, Serbia, Slovakia, Slovenia, Solomon Islands, Vanuatu |
| R5REF | Reforming Ecomonies of the Former Soviet Union | Armenia, Azerbaijan, Belarus, Georgia, Kazakhstan, Kyrgyzstan, Republic of Moldova, Russian Federation, Tajikistan, Turkmenistan, Ukraine, Uzbekistan |

Definitions of IPCC 5-region macro regions as listed in the IPCC AR6 database.