## [Peer Review File · Nature]

Manuscript Title: Aligning climate scenarios to emissions inventories shifts global benchmarks

Reviewer Comments & Author Rebuttals

Reviewer Reports on the Initial Version:

Referees' comments:

Referee #1 (Remarks to the Author):

Review of "Aligning IPCC scenarios to national land emissions inventories shifts global mitigation benchmarks"

Overview:

This paper starts with a significant challenge: we tally emissions (for national accounts reporting for the Paris Accord / stocktake) using inventory techniques that include indirect processes, but we create scenarios for climate model runs from IAMs that keep track of processes and emissions differently (and mostly only account for direct processes). This results in a systematic discrepancy between what think we know from climate models about how land use and land use change impact climate (for example, for critical temperature targets like 2C), and how we might be keeping track of our own progress towards those targets.

The authors then aim to understand the implications for our temperature targets & carbon removal if we were to proceed according to our greenhouse gas inventory techniques. This is an interesting set of analyses, but the paper as it is has some critical shortcomings that need to be addressed. The biggest issue I see is that this analysis builds directly on what was laid out in Grassi et al, Nature Climate Change (2021), but it is not clear that the new findings are either technically correct (see items 2 and 3 below — there are methodological questions, and an overall lack of detail in the presentation that make it difficult to assess), or really solve the fundamental problem (see item 1). Therefore, as a reader, I was left feeling that the work was incremental, and does not rise to the level of importance for publication in Nature (but that is for the editors to decide). I hope my feedback is helpful for the authors as they move forward.

Major Comments:

There are three main areas where the paper needs additional analysis, more careful interpretation, and/or clarity:

(1) What is the solution here?: From Grassi et al (2021) we learned that IAMs have a larger C flux than NGHGs, because national inventories tend to be more aggressive about carbon removals. (That is, the discrepancy is entirely on the carbon removal side of the equation.) It is not totally clear here in the methods whether the same method as in Grassi et al (2021) was applied, though I think so. (Something like Fig 2 in Grassi et al 2021 is sorely needed here.) Isn't the takeaway from the earlier

paper that we should be layering in some sort of DVM or emulator within IAMs to get the correct fluxes to use in ESMs? I am not sure what the new analysis adds here, and in fact some of the lack of clarity (see below) actually makes it seem like the discrepancy is not as big a deal as it really is. My sense is that the deeper issue here is that real carbon flux accounting, especially around removals, is difficult and needs to be both better constrained and (most importantly) better represented in the processes included in IAMs. The global average bias-factor application here is useful for the harmonization described but doesn't really get at the underlying problem in any direct way.

(2) Methodological Uncertainty: The findings on timing and what the accounting gap looks like over time are subject to quite a bit of uncertainty about how the bookkeeping model handles fluxes and apportionments over time. These types of choices are outlined, as one example, in (<https://doi.org/10.1080/17583004.2014.913867>). It is not clear that the one model used here captures any of that uncertainty, and if it did, it might dramatically widen or shift the distributions in Figure 2 (showing the average time shift of 2-3 years).

More generally, there is a real lack of clarity on what was actually done here. The authors point to Ref 4 numerous times, without extracting the main points from that work into this piece. This makes it incredibly hard to follow the paper; I had to read both side by side to understand what was done, and that means this work isn't clear enough to stand on its own. For example, on p.17 - you cite Ref 4, but you never actually state what the adjustment flux is here, even after referring the reader to the methods many times. I think this is pretty straightforward to fix with a careful rewrite, but it this sort of thing does need to be done throughout.

A related concern that the authors note, but do not assess, is the use of one single correction factor globally. The section starting on the bottom of p 17 that describes this reasonably well but the authors could do more to systematically probe this uncertainty (both the magnitude of the factor and the assumed homogeneity).

The authors should also address whether albedo or other changes from the reduced uptake in IAMs also alter the projections? (ie how much of the problem is just in fluxes vs a full treatment of land cover changes). (p. 18 line 10 gives a back of the envelope calculation here for the emissions portion.)

(3) Writing / Presentation. The writing is clear and accurate in a strict sense, but the style is very technocratic and has obfuscated the important points here. A few examples:

Abstract: "appear more ambitious" - unclear which direction this means? Can you rephrase so that it is clear whether the targets or reality are too ambitious?

Introduction: I really appreciate the technical nature of the introduction, but I do think some of the main points are getting lost. I suggest that the authors need a more clear statement about what the actual issue is (e.g. around Line 20, this is not very accessible to a broad audience). I suggest something like: "Climate models use spatially- and temporally- explicit emissions inventories, while NDCs for the Paris Agreement are quite vague in comparison, and in most cases consist of reductions goals with some very coarse sectoral information (e.g., country X will reduce CO2 emissions from the

energy sector by Y% by 2030). As a result, there are important potential discrepancies between the spatio-temporal emissions scenarios utilized for AR6 and how NDCs are likely to be implemented. While this accounting/scenario discrepancy is possible across all sectors, it is likely to be largest in the land use sector.” (This is just suggested wording; I hope the broader point is clear.)

p.3, line 24 - please define the direction of the discrepancy.

p.4 line 5 - You write “We develop a novel approach to estimate large-quantity scenario ensembles of land-based fluxes consistent with national inventories building on previous efforts and provide a translation tool aligning scenarios and national inventories for use in near-term policy making.” After reading the manuscript (and appendix) it is more clear what you mean by this statement, but I think it would help the reader immensely to provide some more descriptive language here. What does it mean to make the scenario-based fluxes consistent with national inventories, and what does it mean to translate and align? You are adding in a missing component and that has to be stated clearly! I think you could consider moving much of the present introduction to discussion, and actually starting the paper with the “Aligning global pathways” section - this is really the heart of the matter!

p.6 line 2 - What does it mean to pass IPCC vetting? - should clarify for broader audience. Especially important - what is the sign of the alignment factor (ie, which relative to which), and what does that mean? I point this out only because the language is rather technical and murky here “alignment gap” etc. I think it would be best to simply state which method is over/under-estimating compared to the other, what processes are likely leading to that mis-estimation or bias, and how it might be fixed in the future.

p. 18, Line 9 - This paragraph is clear and should be in the main manuscript.

Referee #2 (Remarks to the Author):

This seems like a solid analysis of an important and policy-relevant issue: that national inventory estimates of land-related emissions are very much unaligned with scientific estimates of the same quantity. This has far-reaching implications for the overall alignment of national emissions targets with the goals of the Paris agreement, as well as the level of mitigation effort required to meet existing national targets.

My overall assessment of the manuscript is positive, though there are quite a few things about the terminology and methods that are not very clear the way they are currently presented, making it a difficult in some cases to assess the actual quality of the results (see list below).

1) The headline statement of the abstract (“key global mitigation benchmarks ... appear more ambitious”) is not specific enough to be able to interpret what this means. I take it that this means that national targets would need to be made more ambitious in order to achieve the same benchmark? If so, it should be stated in this way (a similar language problem occurs in other places the paper also)

2) The terms "direct" and "indirect" are not intuitive labels to give to the types of carbon fluxes they refer to. In particular "indirect" fluxes implies some causality, which is not the case here -- the fluxes in this category do not result from LULUCF but rather from the effect of climate changes on vegetation processes, so labelling these carbon fluxes as LULUCF indirect fluxes is a misnomer.

3) The use of the term "alignment" is a bit confusing in many places. For example, on page 6, line 8 the phrase "LULUCF emissions aligned with NGHGs" is not at all clear -- I think what is meant is something more like "LULUCF emissions estimated using NGHGI protocols." On the same page (line 4) you refer to an "alignment factor" which is a similarly confusing term -- this one is the difference in emissions estimates between the two methods used, and should be described as such. Similar unclear use of the term alignment needs to be corrected throughout

4) The argument on page 7 for why these results do not imply that the amount of global effort needs to increase is also hard to follow and not very clear. It seems to me that there are significant implications for both national and global effort levels, especially for near-term (2030) targets ... both in terms of the amount of effort needed to achieve the targets, and for how ambitious the targets need to be in order to meet the Paris objectives. For example, if policy-makers believe that land is currently a net sink (contributing to CDR) then they will conclude that less effort is needed to decrease fossil fuel emissions to meet their net-zero targets. Similarly, if a given amount of CDR is required to meet some climate target (according to the IPCC report), this will seem much easier to achieve than it actually is if we consider natural carbon sinks within managed lands to be a contributor to this CDR requirement. And what about offset markets in the corporate world? Offsetting a fossil fuel emission with a natural carbon sink that happens to occur within a managed forest would be very problematic. All of these things seem to suggest that using NGHGI conventions for land accounting leads to an underestimate of the amount of effort that is actually required to meet global temperature goals.

5) phrases like "developed countries see a modest increase in 2030 emissions reductions" on page 8 are not clear ... I take this to mean that developed countries need to adopt more ambitious emissions reductions in order to achieve the same climate effect? This entire section contains ambiguous language whose directionality is difficult to interpret.

6) It is not clear to me how land-based CDR is defined in this study. Figure 4 shows current land-based CDR at 2020 of something like 5.5 GtCO₂ per year, which is not CDR as I understand it ... is this the land-based removal flux that occurs within managed lands (i.e. what is labelled elsewhere as indirect LULUCF fluxes)? If so, this is not CDR by any definition that I have ever encountered, since it is a natural removal (caused by CO₂ fertilization among other things) that happens to occur in managed lands, which is quite different from CDR as usually defined (intentional anthropogenic removal).

Referee #3 (Remarks to the Author):

Review of Gidden et al, "Aligning IPCC scenarios to national land emissions inventories shifts global mitigation benchmarks"

This is a very timely and important paper which will be influential in dealing with a critical issue regarding emissions accounting and setting of national policies towards "net zero"

The paper is clearly intended to come in time to be actively used in the stocktake process - such a methodology is very much needed, and although based on just one model, the science here is robust and will be very useful.

However, precisely because of its importance, it is vital that the message comes across clearly and on first reading I did find the presentation confusing in places and had to reread. To a generalist audience the message needs to be more clear. For example in the abstract the phrase that mitigation benchmarks "appear more ambitious" is not clear to me - does this mean they need to be more ambitious to achieve the goal? Or that they are more ambitious than needed and therefore need to be less ambitious? If the text was more explicit - to a non-expert reader - of the direction of imbalance that would be much easier to read.

As an analogy - imagine you have to make a 10am appointment, but your clock is running slow. The fact your clock is slow does not change the time of the appointment, but it does mean that you may miss it unless you understand the mis-match. In this paper you present such a mis-match but it's not always easy to tell if the clock is fast or slow - which definition of accounting/targets are ahead or behind the other? All this becomes clear of course as you read through - I have no problems with the science of the paper - but it does take multiple reading to get this clarity.

As a more minor issue I missed any discussion of the importance of independent monitoring of reporting. If targets are defined and measured with assumptions of land sink on managed land, then this needs to be monitored - for example from atmospheric measurements precisely to avoid issues that over time the assumptions in the accounting will no longer be valid. There is a stronger than ever role for "top down" MRV and not just "bottom up" reporting.

Minor gripe - the symbols on suppl. figure S1 are too small to easily distinguish - can this be made clearer?

Author Rebuttals to Initial Comments:

Author Response to Reviewer Remarks

We would like to thank each of the three reviewers for their thoughtful and thorough comments on our manuscript. We appreciate that the reviewers recognize the novelty and importance of our contribution to the scientific literature and its impact on global climate policy making. Having implemented the vast majority of reviewer suggestions, we believe our manuscript's quality and clarity have been significantly enhanced.

We have striven to address each comment through targeted edits in the paper and direct responses inline below. Taking the reviewer and editorial comments together, we identified three primary areas where we could improve our manuscript:

1. Clarity of language and concepts: some reviewers highlighted that terms such as indirect, direct, fluxes, removals, and alignment were in some cases unclear and that overall the key process steps for aligning IPCC pathways with national inventories could be better communicated. In addition to making specific adjustments in the text, we have included a new Figure 1 (migrating the old Figure 3 to extended data) to better communicate to readers the fundamental issue that our paper addresses, highlighting key terms and concepts used throughout the paper.
2. Impact on country ambition: all reviewers highlighted that the language around updated benchmarks "appearing more ambitious" was not clear. We agree and have strengthened this and related language in the abstract and main text of our manuscript. The outcome of our analysis is that aggregate country-level targets which include a component of the indirect LULUCF flux need to be made more ambitious to be in line with IPCC-assessed modelled mitigation pathways.
3. The length of the manuscript was slightly too long (by ~10%). To address this while enhancing the readability of the text, we have revised significantly the text to make it more succinct and clear.

Below, we provide directed responses to the individual reviewer comments. The original text of the comments is purple while our responses are in black.

We again thank each of the three referees for their time in reviewing our manuscript and for their suggestions for how to improve upon it.

Referee #1 (Remarks to the Author):

Review of “Aligning IPCC scenarios to national land emissions inventories shifts global mitigation benchmarks”

Overview:

This paper starts with a significant challenge: we tally emissions (for national accounts reporting for the Paris Accord / stocktake) using inventory techniques that include indirect processes, but we create scenarios for climate model runs from IAMs that keep track of processes and emissions differently (and mostly only account for direct processes). This results in a systematic discrepancy between what we think we know from climate models about how land use and land use change impact climate (for example, for critical temperature targets like 2C), and how we might be keeping track of our own progress towards those targets.

The authors then aim to understand the implications for our temperature targets & carbon removal if we were to proceed according to our greenhouse gas inventory techniques. This is an interesting set of analyses, but the paper as it is has some critical shortcomings that need to be addressed. The biggest issue I see is that this analysis builds directly on what was laid out in Grassi et al, Nature Climate Change (2021), but it is not clear that the new findings are either technically correct (see items 2 and 3 below — there are methodological questions, and an overall lack of detail in the presentation that make it difficult to assess), or really solve the fundamental problem (see item 1). Therefore, as a reader, I was left feeling that the work was incremental, and does not rise to the level of importance for publication in Nature (but that is for the editors to decide). I hope my feedback is helpful for the authors as they move forward.

Major Comments:

There are three main areas where the paper needs additional analysis, more careful interpretation, and/or clarity:

(1) What is the solution here?: From Grassi et al (2021) we learned that IAMs have a larger C flux than NGHGs, because national inventories tend to be more aggressive about carbon removals. (That is, the discrepancy is entirely on the carbon removal side of the equation.) It is not totally clear here in the methods whether the same method as in Grassi et al (2021) was applied, though I think so. (Something like Fig 2 in Grassi et al 2021 is sorely needed here.) Isn't the takeaway from the earlier paper that we should be layering in some sort of DVM or emulator within IAMs to get the correct fluxes to use in ESMs? I am not sure what the new analysis adds here, and in fact some of the lack of clarity (see below) actually makes it seem like the discrepancy is not as big a deal as it really is.

The reviewer raises two major questions: what is the solution we propose and more fundamentally what is the issue we are addressing? Our aim is that the new Figure 1 as well as updated text in the beginning of the manuscript elucidate better the answers to both of these questions, addressing the

reviewers concern that such a figure and discussion are 'sorely needed'. However, we also think there may be some slight misunderstandings by the reviewer which we would like to address - we hope that this is useful.

Indeed, Grassi et al (2021) for the first time described the discrepancy between the LULUCF fluxes used in IAMs against those included in national inventory accounting as submitted to the UNFCCC. This discrepancy arises because IAMs consider only "direct" LULUCF fluxes, whereas NGHGs also include some amount of "indirect" human-induced fluxes, which are only considered by physical carbon-cycle models (including the reduced-complexity models used to estimate time evolution of global mean temperature in IAMs). In order to compare like-for-like future estimates of NGHGI-based emissions, one must indeed include an estimation of the future evolution of this "indirect" flux. Grassi et al (2021) provided an estimation for a small subset of scenarios using a DGVM, and one of the key novelties of our study is that we provide an estimation of this flux for all scenarios assessed by the IPCC in AR6 which provide sufficient land-use change data using the reduced complexity carbon-cycle model, OSCAR. This improvement on the state-of-the-art of scenario-based LULUCF flux estimations then enables us to assess how NGHGI-aligned fluxes evolve over time consistent with each IPCC-assessed scenario, thus allowing to update IPCC mitigation benchmark estimations.

Therefore, we do apply the same methodology as Grassi et al. (2021), but using another model that simulates both the direct and indirect terms of the equation (in a probabilistic fashion, see below) for all scenarios available in the IPCC AR6 WG3 database which provide sufficient detail on land use change patterns.

My sense is that the deeper issue here is that real carbon flux accounting, especially around removals, is difficult and needs to be both better constrained and (most importantly) better represented in the processes included in IAMs.

While it is true that LULUCF flux MRV is an area which needs heightened attention in the policy sphere as well as methodological improvements, we focus here on the fundamental mismatch in definitions in terms of which fluxes are accounted for in IAMs compared with NGHGs. Our work tries to bridge this definitional gap. We agree with the reviewer that physical and integrated assessment modelling teams can better coordinate to provide consistent estimates of these fluxes aligned with NGHGs and enhance MRV, which we call for in the conclusion of our manuscript. Our study showcases one way that these communities can begin to do so.

The global average bias-factor application here is useful for the harmonization described but doesn't really get at the underlying problem in any direct way.

The ‘bias-factor’ as referenced here is in fact a regionally assessed, temporally evolving component of the overall indirect land carbon flux (which we call the ‘adjustment factor’, see, e.g., the updated Fig 1.), computed with OSCAR as described in the methods section of our manuscript. Therefore, it is neither global nor averaged. We have updated our manuscript to highlight this issue specifically:

To estimate the direct and indirect components of land-based carbon fluxes necessary to align mitigation pathways with conventions used in NGHGs, we use a reduced-complexity climate model with explicit treatment of the land-use sector, OSCAR⁵, one of the models used for the annual Global Carbon Budget³, applied in a probabilistic setup and at a resolution of five global regions used in IPCC assessments. We calculate the discrepancy between model and NGHGI-based accounting methods globally to be $4.4 \pm 1.0 \text{ Gt CO}_2 \text{ yr}^{-1}$ averaged over the 2000-2020 time period, which is in line with existing estimates^{2,5}. We then assess the pathways with OSCAR to quantify how this gap evolves over time.

We compute this flux consistent with the land-use patterns at the regional level reported for each scenario. This flux is not dependent on anthropogenic land use mitigation decisions (which are all captured by the direct LULUCF flux which is included in the decision-making frameworks within IAMs), and hence needs to be computed outside of IAMs, as we do in our manuscript, in order to make the link between modelled pathways and national inventories. We believe this is one way, among others, to get directly at “the underlying problem”.

(2) Methodological Uncertainty: The findings on timing and what the accounting gap looks like over time are subject to quite a bit of uncertainty about how the bookkeeping model handles fluxes and apportions them over time. These types of choices are outlined, as one example, in (<https://doi.org/10.1080/17583004.2014.913867>). It is not clear that the one model used here captures any of that uncertainty, and if it did, it might dramatically widen or shift the distributions in Figure 2 (showing the average time shift of 2-3 years).

The reviewer is correct that timing is uncertain because of apportionment of carbon fluxes. This uncertainty, however, is fully sampled through the probabilistic setup used to run OSCAR. Details on the OSCAR setup are provided in Methods and references therein (esp. Gasser et al., 2020; ref. 5). In short, turnover times and allocation coefficients of the model’s carbon pools are taken from a prior distribution of parameters derived from state-of-the-art DGVMs. The output of the Monte Carlo simulations are then constrained ex-post to match reference assessments of historical fluxes, so that the final average timing is a result of both the prior parameter distributions and this constraining. We added mention of the probabilistic framework in two places in the main text to clarify this point, both as cited above as well as:

We provide in this work a novel estimation of LULUCF emissions consistent with NGHGs, for all IPCC-assessed scenarios which provide sufficient information to do so, using probabilistic and constrained estimates from a single established model.

More generally, there is a real lack of clarity on what was actually done here. The authors point to Ref 4 numerous times, without extracting the main points from that work into this piece. This makes it incredibly hard to follow the paper; I had to read both side by side to understand what was done, and that means this work isn't clear enough to stand on its own. For example, on p.17 - you cite Ref 4, but you never actually state what the adjustment flux is here, even after referring the reader to the methods many times. I think this is pretty straightforward to fix with a careful rewrite, but it this sort of thing does need to be done throughout.

We appreciate the comment by Reviewer 1. We have struggled to include more methodological details in the main manuscript while also communicating in a sufficient manner our key results. As a compromise, we have decided to develop a key overview figure which now is featured prominently as Figure 1. Our aim is that this figure provides a clear overview of the complexities of the alignment process used and in fact simplifies its description.

In addition to the figure, we have further clarified the method text as follows:

Following the approach by ref⁴, the adjustment flux (i.e., the indirect LULUCF flux included in NGHGs but not included by IAMs, also called the 'factor in main text') required to move from bookkeeping emissions to NGHGI-compatibles ones is calculated as the part of the land carbon sink that occurs in forests that are managed.

In terms of the adjustment flux, we do provide these regional and global timeseries in the supplementary data (provided to reviewers via the IIASA Scenario Explorer link), which will be published open access upon publication of this manuscript. We provide graphically this flux estimate in (original) Figure 1, though we agree this could be stated more explicitly. To address this, we now provide in the supplemental information a new table (Extended Data Table 1) with our estimates of the additional flux both globally and at the regional resolution of our analysis.

A related concern that the authors note, but do not assess, is the use of one single correction factor globally. The section starting on the bottom of p 17 that describes this reasonably well but the authors could do more to systematically probe this uncertainty (both the magnitude of the factor and the assumed homogeneity).

We do not use a single factor globally as described above. The key limitation of our study, which we recognize and state, is that we use only a single model to make this estimation and further improvements on our approach laid out for the first time in this paper could include a multi-model comparison of this flux estimation. We additionally provide uncertainty assessments in multiple key locations, e.g., in (original) Figures 1 and 4, where we report both scenario uncertainty as well as carbon cycle uncertainty where appropriate.

The authors should also address whether albedo or other changes from the reduced uptake in IAMs also alter the projections? (ie how much of the problem is just in fluxes vs a full treatment of land cover changes). (p. 18 line 10 gives a back of the envelope calculation here for the emissions portion.)

Albedo change is not a part of the issue we discuss in this paper. The radiative forcing and ensuing climate change induced by it were estimated and included in the scenario projections of the AR6 database. We added a sentence in Methods to clarify this:

To run the final scenario simulations over 2014-2100, OSCAR needs two types of input data: CO₂ and local climate projections, and land use and land cover change projections. The former mostly affect the land carbon sink (i.e. the indirect effect), while the latter mostly affect the bookkeeping emissions (i.e. the direct effect). OSCAR follows a theoretical framework⁴¹ that enables clear separation of both direct and indirect effects. Only the direct effect is reported annually in the GCB. Note that we do not reevaluate the land cover change albedo effect since this was already included in the original AR6 database climate projections.

(3) Writing / Presentation. The writing is clear and accurate in a strict sense, but the style is very technocratic and has obfuscated the important points here. A few examples:

Abstract: “appear more ambitious” - unclear which direction this means? Can you rephrase so that it is clear whether the targets or reality are too ambitious?

We agree with the reviewer that clarity is needed here and have addressed this in our top-level responses.

Introduction: I really appreciate the technical nature of the introduction, but I do think some of the main points are getting lost. I suggest that the authors need a more clear statement about what the actual issue is (e.g. around Line 20, this is not very accessible to a broad audience). I suggest something like: “Climate models use spatially- and temporally- explicit emissions inventories, while NDCs for the Paris Agreement are quite vague in comparison, and in most cases consist of reductions

goals with some very coarse sectoral information (e.g., country X will reduce CO2 emissions from the energy sector by Y% by 2030). As a result, there are important potential discrepancies between the spatio-temporal emissions scenarios utilized for AR6 and how NDCs are likely to be implemented. While this accounting/scenario discrepancy is possible across all sectors, it is likely to be largest in the land use sector.” (This is just suggested wording; I hope the broader point is clear.)

We thank the reviewer for their suggestion, but while the reviewer’s text is factually correct, this discrepancy in the quality and details of the inventories vs. NDCs data is not the focus of our paper. The crux of our work is a definitional inconsistency that we quantify to allow like-for-like comparison of IPCC scenarios and NDCs.

At the same time, we have significantly shortened the introduction and have tried to streamline it to better introduce the main issue we address in the paper, including as well a new conceptual Figure 1. We hope these changes help address the reviewer’s point about not losing the main points of the paper.

p.3, line 24 - please define the direction of the discrepancy.

We have now clarified the directionality of the discrepancy:

A key discrepancy exists, however, in how model-based scientific studies and national greenhouse gas inventories (NGHGs) account for the role of anthropogenic land-based carbon fluxes^{4,15,16}, with national inventories incorporating a broader scope of removals², resulting in lower emission estimates in NGHGs.

p.4 line 5 - You write “We develop a novel approach to estimate large-quantity scenario ensembles of land-based fluxes consistent with national inventories building on previous efforts and provide a translation tool aligning scenarios and national inventories for use in near-term policy making.” After reading the manuscript (and appendix) it is more clear what you mean by this statement, but I think it would help the reader immensely to provide some more descriptive language here. What does it mean to make the scenario-based fluxes consistent with national inventories, and what does it mean to translate and align? You are adding in a missing component and that has to be stated clearly! I think you could consider moving much of the present introduction to discussion, and actually starting the paper with the “Aligning global pathways” section - this is really the heart of the matter!

We thank the reviewer for their suggestion. We have now completely removed this and the following paragraph so that the “Aligning global pathways” section comes immediately after a

statement of the problem at hand. That section now begins with describing what it means to make scenario-based fluxes consistent with national inventories, including with a new conceptual figure to help the reader synthesize these complex concepts.

p.6 line 2 - What does it mean to pass IPCC vetting? - should clarify for broader audience. Especially important - what is the sign of the alignment factor (ie, which relative to which), and what does that mean? I point this out only because the language is rather technical and murky here "alignment gap" etc. I think it would be best to simply state which method is over/under-estimating compared to the other, what processes are likely leading to that mis-estimation or bias, and how it might be fixed in the future.

We have completely refactored this section to enhance language clarity especially around the alignment factor. It now reads as:

We calculate 4.4 ± 1.0 Gt CO₂ yr⁻¹ higher LULUCF emissions globally averaged over the 2000-2020 time period using model-based accounting conventions compared to NGHGI-based accounting, which is in line with existing estimates^{2,5}. We then assess the pathways with OSCAR to quantify how this gap evolves over time. A total of 914 of the 1202 IPCC-assessed pathways provided sufficient land-use change data to enable this alignment. A full description of the calculation approach is provided in the Methods.

p. 18, Line 9 - This paragraph is clear and should be in the main manuscript.

We have had to make difficult decisions regarding manuscript length and removal of significant portions of pre-existing main text, and would prefer to keep this paragraph in the Methods section.

Referee #2 (Remarks to the Author):

This seems like a solid analysis of an important and policy-relevant issue: that national inventory estimates of land-related emissions are very much unaligned with scientific estimates of the same quantity. This has far-reaching implications for the overall alignment of national emissions targets with the goals of the Paris agreement, as well as the level of mitigation effort required to meet existing national targets.

My overall assessment of the manuscript is positive, though there are quite a few things about the terminology and methods that are not very clear the way they are currently presented, making it a difficult in some cases to assess the actual quality of the results (see list below).

1) The headline statement of the abstract ("key global mitigation benchmarks ... appear more ambitious") is not specific enough to be able to interpret what this means. I take it that this means that national targets would need to be made more ambitious in order to achieve the same

benchmark? If so, it should be stated in this way (a similar language problem occurs in other places the paper also)

We agree with the reviewer that clarity is needed here and have addressed this in our top-level responses.

2) The terms "direct" and "indirect" are not intuitive labels to give to the types of carbon fluxes they refer to. In particular "indirect" fluxes implies some causality, which is not the case here -- the fluxes in this category do not result from LULUCF but rather from the effect of climate changes on vegetation processes, so labelling these carbon fluxes as LULUCF indirect fluxes is a misnomer.

As much as we agree with the fundamental point made by the reviewer, the issue of which components of the net land-to-atmosphere carbon flux should be considered anthropogenic is exactly the one our paper is dealing with. The reviewer might consider that the indirect flux should not be labelled LULUCF (which, again, we agree with) but national inventories do consider it anthropogenic if it occurs over managed land! Hence, the discrepancy between inventories and scientific assessments. Therefore, when we label this flux as anthropogenic, we only follow the inventories' logic.

Regarding the use of the direct/indirect terminology, we have settled on this because we think it remains the simplest and most straightforward one. We respectfully disagree with the reviewer that there is no causality in the indirect term: the CO₂-fertilisation effect (and response to climate change) is caused by elevated CO₂ that is itself caused by anthropogenic emissions (hence, the "indirect" epithet) – indeed, this indirect flux is explicitly defined as anthropogenic in the IPCC guidelines used by UNFCCC parties to account for their LULUCF emissions (i.e., the NGHGI convention) as opposed to natural fluxes due to, e.g., interannual variability. Alternatives used in the literature would be: land use change emissions = direct flux = anthropogenic flux = bookkeeping flux; as well as: land sink = indirect flux = natural flux = environmental flux. Of these, we cannot use anthropogenic/natural because of the issue the paper deals with (that inventories and assessments do not use the same definition of "anthropogenic"), we cannot use emissions/sink because of the potential reversal of those fluxes (LUC emissions become a sink, and the land sink becomes a source), and we find "bookkeeping" too technical and "environmental" too vague. Therefore, we kept our original direct/indirect terminology, but also clarified in text their definition.

In any case, we thank the reviewer for raising the important point of being precise with terminology, and hope that our explanation here helps them to understand how we have landed on these exact terms (after much internal debate).

3) The use of the term "alignment" is a bit confusing in many places. For example, on page 6, line 8

the phrase "LULUCF emissions aligned with NGHGs" is not at all clear -- I think what is meant is something more like "LULUCF emissions estimated using NGHGI protocols." On the same page (line 4) you refer to an "alignment factor" which is a similarly confusing term -- this one is the difference in emissions estimates between the two methods used, and should be described as such. Similar unclear use of the term alignment needs to be corrected throughout

We have tried to sharpen specific points around terminology with respect to 'alignment' in our manuscript, while at the same time recognizing that we need to write for a wide audience. Indeed, the reviewer is correct that by 'alignment' we mean 'assessing LULUCF emissions fluxes using NGHGI conventions', which we can not of course repeat every time the concept arises. Instead, we now introduce this concept early in the manuscript:

Aligning mitigation pathways assessed by the IPCC with emissions inventory conventions is therefore needed to support science-based formulation of new NDCs and to measure collective global action against emission levels necessary to achieve the Paris Agreement.

We again highlight this definition when first introducing the difference in our estimation between both conventions, as requested by the reviewer:

Across both 1.5°C and 2°C scenarios (Fig. 2A, definitions in Methods), LULUCF emissions estimated using NGHGI conventions show a strong increase in the total land sink until around mid-century. However, the 'NGHGI alignment gap' (Fig. 2B) decreases over this period, nearing zero in the 2050-2060s for 1.5°C scenarios and 2070s-2080s for 2°C scenarios.

And we have striven to make sure our use of the term is clear throughout. However, we still use the term, as we think it is critical to provide a concise word describing the overall concept we are trying to convey.

4) The argument on page 7 for why these results do not imply that the amount of global effort needs to increase is also hard to follow and not very clear. It seems to me that there are significant implications for both national and global effort levels, especially for near-term (2030) targets ... both in terms of the amount of effort needed to achieve the targets, and for how ambitious the targets need to be in order to meet the Paris objectives. For example, if policy-makers believe that land is currently a net sink (contributing to CDR) then they will conclude that less effort is needed to decrease fossil fuel emissions to meet their net-zero targets. Similarly, is a given amount of CDR required to meet some climate target (according to the IPCC report), this will seem much easier to achieve than it actually is if we consider natural carbon sinks within managed lands to be a contributor to this CDR requirement. And what about offset markets in the corporate world?

Offsetting a fossil fuel emission with a natural carbon sink that happens to occur within a managed forest would be very problematic. All of these things seem to suggest that using NGHGI conventions for land accounting leads to an underestimate of the amount of effort that is actually required to meet global temperature goals.

We completely agree with the reviewer and thank them for raising this point. We have addressed the issue around 'implications of global effort' in our top-line comments. Additionally, we have now elevated these concerns throughout the manuscript (e.g., highlighting "*the risk of overdependence on land-sinks to measuring mitigation progress using national inventory conventions*" and "*the future effort needed to achieve or maintain net-zero economy-wide emissions would be underestimated using NGHGI accounting conventions as the indirect contribution to land sinks lose efficacy and eventually become a net source of emissions in low-warming scenarios.*").

5) phrases like "developed countries see a modest increase in 2030 emissions reductions" on page 8 are not clear ... I take this to mean that developed countries need to adopt more ambitious emissions reductions in order to achieve the same climate effect? This entire section contains ambiguous language whose directionality is difficult to interpret.

We agree with the reviewer that our use of language here was slightly ambiguous and have tried to clarify. We have rewritten significant portions of this paragraph, and in particular have tried to address the raised issue with the following revised text:

Globally, the relative difference in emissions reduced between 2020-2030 in 1.5°C pathways between the two accounting frameworks is small. Regionally, though, 1.5°C-consistent emission reductions are higher for developed countries, whereas needed emission reductions in most developing regions are slightly lower when assessing scenario outcomes using NGHGI-based conventions.

6) It is not clear to me how land-based CDR is defined in this study. Figure 4 shows current land-based CDR at 2020 of something like 5.5 GtCO₂ per year, which is not CDR as I understand it ... is this the land-based removal flux that occurs within managed lands (i.e. what is labelled elsewhere as indirect LULUCF fluxes)? If so, this is not CDR by any definition that I have ever encountered, since it is a natural removal (caused by CO₂ fertilization among other things) that happens to occur in managed lands, which is quite different from CDR as usually defined (intentional anthropogenic removal).

The reviewer raises a critically important definitional point. Indeed CDR in the context of the IPCC can be considered to be the negative component of the land-related carbon flux on managed land. As explained in the Methods, this is taken as the net flux over the "Forest" land category as defined by Grassi et al. (2022) in ref 2, corresponding to carbon which is captured and durably stored in land-

based reservoirs, consistent with the common definition of CDR as adopted by the IPCC. We agree that to the best of our knowledge, the IPCC definition corresponds only to direct-flux-based removals, and thus have updated Figs. 4A and 4B to reflect this, as well as the computed Land CDR numbers in the text.

It is important to note that estimating this value poses similar challenges with respect to different conventions between models and NGHGI in the context of the UNFCCC. We also report in Figs. 4C and 4D the negative component of the flux on land considered managed by NGHGI conventions but not by model conventions (labelling it Indirect removals). This distinction is critical because of how removals may be measured, reported, and verified in the context of future voluntary and compliance carbon markets. We have striven, though, to separate the two concepts by calling the direct component Land CDR (Direct) and the indirect component Indirect Removals, specifically eschewing the CDR moniker.

Referee #3 (Remarks to the Author):

Review of Gidden et al, "Aligning IPCC scenarios to national land emissions inventories shifts global mitigation benchmarks"

This is a very timely and important paper which will be influential in dealing with a critical issue regarding emissions accounting and setting of national policies towards "net zero"

The paper is clearly intended to come in time to be actively used in the stocktake process - such a methodology is very much needed, and although based on just one model, the science here is robust and will be very useful.

However, precisely because of its importance, it is vital that the message comes across clearly and on first reading I did find the presentation confusing in places and had to reread. To a generalist audience the message needs to be more clear. For example in the abstract the phrase that mitigation benchmarks "appear more ambitious" is not clear to me - does this mean they need to be more ambitious to achieve the goal? Or that they are more ambitious than needed and therefore need to be less ambitious? If the text was more explicit - to a non-expert reader - of the direction of imbalance that would be much easier to read.

We agree with the reviewer that clarity is needed here and have addressed this in our top-level responses. Throughout the manuscript we have provided clearer outcomes and messaging related to how mitigation benchmarks are affected when translating from model-based LULUCF accounting conventions compared to NGHGI-based conventions. Where useful, we have striven to note the direction of change more explicitly as also pointed out by other reviewers.

As an analogy - imagine you have to make a 10am appointment, but your clock is running slow. The

fact your clock is slow does not change the time of the appointment, but it does mean that you may miss it unless you understand the mis-match. In this paper you present such a mis-match but it's not always easy to tell if the clock is fast or slow - which definition of accounting/targets are ahead or behind the other? All this becomes clear of course as you read through - I have no problems with the science of the paper - but it does take multiple reading to get this clarity.

We agree with the reviewer and have tried to address this in the new Figure 1. We have also revised the text overall to reiterate the directionality of the shift in emissions in the near and long-term. To also support this discussion, we have added to (old) Figure 2 (new Figure 3) a plot of emission reduction changes.

As a more minor issue I missed any discussion of the importance of independent monitoring of reporting. If targets are defined and measured with assumptions of land sink on managed land, then this needs to be monitored - for example from atmospheric measurements precisely to avoid issues that over time the assumptions in the accounting will no longer be valid. There is a stronger than ever role for "top down" MRV and not just "bottom up" reporting.

We agree that it is worthwhile to highlight issues around MRV, which is an important and timely topic to consider especially in the context of the GST and future NDC revisions. We have struggled to limit the text of the manuscript, and do not believe we have sufficient space to provide a robust discussion on MRV issues here, but we do try to highlight it in multiple locations.

First in the main text section on 'Balancing Practicalities with Policy Guidance' we discuss as follows:

Understanding and addressing how these different frameworks can be mutually interpreted is a fundamental challenge for evaluating progress towards the Paris Agreement, given the reality that carbon removals from anthropogenic and natural land-based processes cannot be estimated separately by NGHGs, which are typically based on direct observations. The outcomes presented here highlight that the conventions by which land-based carbon removals are considered have important implications for NDC assessment and transparency, operationalization of removals under carbon markets as laid out in Article 6.4 of the Paris Agreement, and monitoring, reporting, and verification (MRV) of these removals.

We have included an additional recommendation in the conclusion of the paper with the following text:

The policy and scientific communities can take steps to meet this challenge by reconciling terms, definitions, and estimates of land-based CO₂ fluxes in four concrete ways. First, targets can be formulated explicitly for areas of critical mitigation action, including gross CO₂ emission reductions without LULUCF, net land-based removals, engineered carbon removals, and non-CO₂ GHG reductions, allowing for parties to clearly define their expected contributions and to measure progress in each domain separately. While most targets are intentionally vague to allow parties to develop bespoke, flexible mitigation approaches based on their respective capabilities and national circumstances, providing additional clarity is critical to enable translation between modelled pathways and aggregate national targets. Second, parties can clarify the nature of their deforestation pledges, since direct and indirect carbon fluxes vary greatly in different forest types³³. Third, scientific and practitioner communities can convene discussions on how to improve MRV of LULUCF fluxes to better align estimates from both groups. Fourth, IAM teams can provide their individual assumptions and estimates for direct LULUCF emissions and removals, indirect fluxes included in NGHGs (as some teams have started doing³⁴), and their assumptions about the land-use contribution of NDCs and LTSs as part of their standard output. Future IPCC assessments could use such data to consider whether scenarios provide sufficient information to inform assessments of necessary collective action aligned with accounting practices used by parties to the UNFCCC. It is critical that such changes be made as part of a community effort to avoid double counting emissions reductions due to realignment to NGHGs.

Minor gripe - the symbols on suppl. figure S1 are too small to easily distinguish - can this be made clearer?

We have enhanced the visual quality of figure S1 in response to this comment. We hope that it is now clearer to read.

Chris Jones

Reviewer Reports on the First Revision:

Referees' comments:

Referee #1 (Remarks to the Author):

Review of 443312

I appreciate the work the authors have undertaken to improve the clarity of the manuscript. The changes are helpful, but there is still a lack of clarity throughout.

The core issue is this: Because you use the language “aligning” and the directionality of language shifts throughout, it is still confusing as written. I still believe there needs to be a clear set of logical steps outlined for readers to understand the analysis flow, and the language and signs have to be 100% consistent and reinforced throughout. The rewritten manuscript is better at clearly stating that IAMs/scenarios don't account for indirect fluxes and therefore don't correspond well to NGHGs (the main policy metric). But the reader then needs to understand that you want to make model output more correct to correspond to the NGHGs, and you will do this by incorporating indirect emissions (OSCAR), and then introducing a purely empirical tuning parameter (“alignment factor”) to make the modeled emissions match the NGHGs. By building the scientific infrastructure for understanding changes over time in the same framework as NGHGs, you are filling an important short-term gap for the GST.

Although I like the idea behind Figure 1, it's not really coming across as intended. For example, it's not obvious that the adjustment that you calculate, from the OSCAR results, is constrained by the NGHGs. Like if Red+Blue=Green, I think you could be more direct that you start with Green and Red, but then calculate the Indirect and Alignment Factor components. I also think that the alignment parameter (gap), since it's essentially a tuning parameter, should be put in a different color? Or, in the “Scientific modeling convention” row, you could break the blue blocks into a blue + alignment factor box? In part (b), it's not clear that it's a 2x2, or how one can best read the figure. Some additional text could help, indicating the same direct fluxes in each column. And again, having the alignment factor separate (or somehow harmoniously illustrated) would again be helpful.

Finally, I still don't see any discussion of how OSCAR might compare to other options that could have been used. It would be helpful to have at least a few sentences about this. What is OSCAR good at/less good at? Maybe the way to think about it is this: if the UNFCCC decided this was the methodology to use, and to implement a ‘scenario translator’ to do this, what would you be nervous about?

Related, this is an important topic, but I'm still left a bit confused by the implications, particularly insofar as they interact with the GST (a main motivator for this work). Isn't there room for some sharper recommendations? Especially: this is the ex-post version of the LULUCF adjustment; is there a way to incorporate this into the actual scenario development infrastructure for (eg) AR7? (Ultimately, don't we want to *not* have to undertake this alignment process in the future? How do we get there?)

ED Figure 1 - Here's an example of where the language could be better - "Original IPCC Benchmark from IAMs" and (I suggest) "New IPCC Benchmark from Improved LULUCF modeling"

ED Figure 2 - the 2C curve and the Current policies point look like they are the same color; this is a bit confusing.

ED Figure 3 - again, I think you could be more clear here where you discuss this - Improved LULUCF modeling modifies the 2030 emissions reductions required differently for 1.5C versus 2C. (The use of "gap" here is also potentially confusing for a reader.)

ED Figure 7 - K-S Tests that look like they reject the null should be discussed very briefly in the caption.

Small nitpicky thing - you refer to IAM and DVM output as "scientific models" in comparison to the NGHGI values, but those inventory values are derived from the scientific literature. Just another place where the language could be careful and consistent.

Figure 2 - I think you could adjust the legends a bit here (and in the corresponding text) for clarity. For example, clearly state "A positive gap means that IAMs overestimated net LULUCF emissions.

I like the plots in the ED Figure 1 versus the distributions in Figure 3, but I think you could combine them (two columns) for a very effective figure?

Referee #2 (Remarks to the Author):

I am generally satisfied that the authors have addressed by previous comments. I have a two follow-up comments, and two small but important new ones.

1) I still find that the language of the abstract is unclear and does not well communicate the directionality and significance of the findings. I would suggest the following text in place of lines 28-35:

"When calculated using NGHGI conventions, key global mitigation benchmarks become harder to achieve, requiring both earlier net zero CO₂ timing and lower cumulative emissions. Furthermore weakening natural carbon removal processes have the potential to mask anthropogenic land-based removal efforts, with the result that land-based carbon fluxes in NGHGIs may ultimately become sources of emissions by 2100. Our results are of critical importance to the Global Stocktake, suggesting that many nations will need to increase the ambition of their climate targets to remain consistent with global temperature goals."

2) I do agree with the authors' argument that the term "indirect" is a reasonable one to apply to natural carbon sinks. However, my quibble is that only a very small portion of this flux is an indirect response to LULUCF itself — rather it is an indirect land carbon flux that is driven by mostly (>90%)

fossil fuel emissions. So I still think the phrase “indirect LULUCF flux” implies causation from land management that is not the case.

3) Figure 1 — I like this new figure, but please do not use the term “virgin land” (this is quite a problematic term that carries misogynistic undertones) ... I would suggest maybe just “Other land” here instead.

4) Finally it seems important to acknowledge either in Figure 1 or elsewhere in the manuscript that the terms “managed” and “unmanaged land” generally only refer to industrial management by mainstream economic activity. However, most of not all of what you refer to as “unmanaged” has in fact been sustainably managed by indigenous communities who have lived there for millennia.

Referee #3 (Remarks to the Author):

Thanks to the authors for taking serious time and effort to address all 3 (positive) reviews of this MS. The result is looking very good. It seemed pretty clear that most of the comments were around the presentation, and very little doubt that the science is robust and important. So I hope this can move towards acceptance and publication. I have a few comments though which hopefully will further improve clarity.

This is one of those issues which takes a while to fully comprehend. For me at least I have to think it through a couple of times to be clear. Therefore, it is very difficult for the authors, who are experts and already have this clearly in their mind, to see the stumbling blocks to clarity. Fig 1 really helps actually, but I found bits of the text still require having to stop and think through which way the “more” or “less” ambitious phrasing means.

For example, line 105; the “LULUCF convention results in more ambitious benchmarks...”. This could be read as “when countries use the LULUCF convention they set themselves targets/benchmarks which are more ambitious than if they had used a model-based convention” (i.e. they can actually relax their targets somewhat), or it could be read as “when countries rely on LULUCF convention to assess their progress towards benchmarks they need to make the benchmarks more ambitious” (i.e. they need to work harder to reduce emissions). The correct reading is the latter, but the reader shouldn’t have to stop and think it through.

It might help to try to see this from the point of view of a new reader and spell it out much more simply than you might imagine is needed.

- The issue is that there are two ways of measuring land use emissions at national scale
- If countries set a target based on modelling (e.g. as per IPCC timelines for the need for net-zero), but then monitor their progress based on accounting (which is what they should be reporting) they will miss their target (by emitting too much)
- They therefore need to make the targets they set themselves more ambitious
 - o Emissions they report need to be lower in their reporting than the modelling would suggest
 - o Date of net zero they aim for needs to be earlier than modelling would suggest

When this is implemented in the text it might not actually need much of a change. E.g. this sentence

(line 105):

“...LULUCF conventions results in more ambitious emissions benchmarks...”

could become:

“...LULUCF conventions results in the need to set more ambitious emissions benchmarks...”

Similarly, the sentence on line 109 that “emissions reductions ... are enhanced” could be mis-read as using LULUCF accounting is better because you end up with greater emissions reductions. Whereas it means that in order to meet targets, emissions reductions measured using LULUCF accounting must be greater than model-based numbers. So maybe change:

“Emissions reductions in 2030 relative to 2020 are enhanced by 3-6 percentage points for both pathway categories”

to

“Emissions reductions in 2030 relative to 2020 need to be 3-6 percentage points greater for both pathway categories if assessed using LULUCF accounting”

Apologies for a long-winded discussion on what will probably sound pedantic to you – but I think it will be vital for the success of the paper to be crystal clear to a new reader. As an aside, this whole issue is similar to the choice of CO₂-equivalence metric used to define net-zero GHG. IPCC WG1 has a horrible description (WG1 SPM, para D.1.8) of this which is so convoluted I don't think even the authors understand it 😊. It would be great if your paper can avoid also having technically-correct but near impossible-to-interpret text.

Regarding figure 1, my comment here might be problematic – maybe an issue the editor can contribute to – it feels like 1A is really useful up front as an explainer of the basic issue being addressed. But panel 1B feels like it should come after the results are shown – it almost pre-empts what the results are. In a perfect world I would recommend splitting the figure, and having 1B come after figure 2 at least. It would be useful during the discussion section to have 1B there. But I realise this increases the total number of figures, even if the total number of panels is unchanged. If the journal would be willing to accept that then it could help the flow. If that is not allowed, then I would be OK to keep them together as one figure.

Minor points

- Can you clarify what “land-based CDR” refers to (line 166) – from figure 4 it doesn't include BECCS right? So what you mean here is CDR which relies on land-based storage, rather than land-based removal from the atmosphere?

- Paragraph from line 181 – if you want a (quantitative) reference that future mitigation scenarios reduce or even reverse the land-sink before the end of this century, then IPCC WG1 has a figure on this – chapter 4, fig 4.7 shows the land sink under the SSPs, and SSP1-1.9 sees a net negative “natural” land sink from around 2070s onwards. So from this point in a roughly 1.5-degree scenario the LULUCF accounting convention will actively work against countries ability to maintain net zero. This gives good consistency and support to your results in figure 2

- make figure 2 the same colours as figure 1 – in 1A, green is NGHGI and red/blue are models, but in figure 2 these are swapped!

- Figure 3 – related to comments on text clarity, maybe the figure could have the zero line more prominent and label above it as “LULUCF-convention defines NZ earlier” for positive values, and

“model conventions define NZ earlier” for negative values. Again, this makes it super clear and unambiguous for a reader not as familiar with it as the authors

Author Rebuttals to First Revision:

Author Response to Reviewer Remarks

We would again like to thank each of the three reviewers for their thoughtful and thorough comments on our manuscript.

We have done our utmost to respond to each suggestion. The original text is in black and our responses are in blue.

Referees' comments:

Referee #1 (Remarks to the Author):

Review of 443312

I appreciate the work the authors have undertaken to improve the clarity of the manuscript. The changes are helpful, but there is still a lack of clarity throughout.

The core issue is this: Because you use the language “aligning” and the directionality of language shifts throughout, it is still confusing as written. I still believe there needs to be a clear set of logical steps outlined for readers to understand the analysis flow, and the language and signs have to be 100% consistent and reinforced throughout. The rewritten manuscript is better at clearly stating that IAMs/scenarios don’t account for indirect fluxes and therefore don’t correspond well to NGHGs (the main policy metric). But the reader then needs to understand that you want to make model output more correct to correspond to the NGHGs, and you will do this by incorporating indirect emissions (OSCAR), and then introducing a purely empirical tuning parameter (“alignment factor”) to make the modeled emissions match the NGHGs. By building the scientific infrastructure for understanding changes over time in the same framework as NGHGs, you are filling an important short-term gap for the GST.

We have adjusted text to try to clarify this concept throughout the manuscript. Figure 2 has been adapted to enhance clarity of the adjustment process (red + blue = green) and directionality of shift (Figure 2C).

Although I like the idea behind Figure 1, it’s not really coming across as intended. For example, it’s not obvious that the adjustment that you calculate, from the OSCAR results, is constrained by the NGHGs. Like if Red+Blue=Green, I think you could be more direct that you start with Green and Red, but then calculate the Indirect and Alignment Factor components. I also think that the alignment parameter (gap), since it’s essentially a tuning parameter, should be put in a different color? Or, in the “Scientific modeling convention” row, you could break the blue blocks into a blue + alignment factor box? In part (b), it’s not clear that it’s a 2x2, or how one can best read the figure. Some additional text could help, indicating the same direct fluxes in each column. And again, having the alignment factor separate (or somehow harmoniously illustrated) would again be helpful.

The blue factor is precisely what is computed by OSCAR in order to align to the NGHGs. It is the indirect component of the LULUCF flux on land considered managed by NGHGs, which constitutes the alignment factor. We hope updates also to Figure 2 help clarify.

Finally, I still don't see any discussion of how OSCAR might compare to other options that could have been used. It would be helpful to have at least a few sentences about this. What is OSCAR good at/less good at? Maybe the way to think about it is this: if the UNFCCC decided this was the methodology to use, and to implement a 'scenario translator' to do this, what would you be nervous about?

We have added text in the discussion section on how to improve on our approach (using multiple models), recommendations on how to incorporate our approach in future assessments, and technical limitations specifically with respect to OSCAR in the methods.

Related, this is an important topic, but I'm still left a bit confused by the implications, particularly insofar as they interact with the GST (a main motivator for this work). Isn't there room for some sharper recommendations? Especially: this is the ex-post version of the LULUCF adjustment; is there a way to incorporate this into the actual scenario development infrastructure for (eg) AR7? (Ultimately, don't we want to *not* have to undertake this alignment process in the future? How do we get there?)

We make this recommendation already, and now include that this process can be used by AR7 scenario database infrastructure if it is missing.

ED Figure 1 - Here's an example of where the language could be better - "Original IPCC Benchmark from IAMs" and (I suggest) "New IPCC Benchmark from Improved LULUCF modeling"

ED Figure 2 - the 2C curve and the Current policies point look like they are the same color; this is a bit confusing.

This has been updated

ED Figure 3 - again, I think you could be more clear here where you discuss this - Improved LULUCF modeling modifies the 2030 emissions reductions required differently for 1.5C versus 2C. (The use of "gap" here is also potentially confusing for a reader.)

The gap language has been updated throughout the manuscript now to only refer to the emissions gap (e.g., in the UNEP Emission Gap report).

ED Figure 7 - K-S Tests that look like they reject the null should be discussed very briefly in the caption.

Language has been added.

Small nitpicky thing - you refer to IAM and DVM output as “scientific models” in comparison to the NGHGI values, but those inventory values are derived from the scientific literature. Just another place where the language could be careful and consistent.

Figure 2 - I think you could adjust the legends a bit here (and in the corresponding text) for clarity. For example, clearly state “A positive gap means that IAMs overestimated net LULUCF emissions.

This has now been added.

I like the plots in the ED Figure 1 versus the distributions in Figure 3, but I think you could combine them (two columns) for a very effective figure?

While we like this idea, this was our initial version of the figure and it made the plot too busy.

Referee #2 (Remarks to the Author):

I am generally satisfied that the authors have addressed by previous comments. I have a two follow-up comments, and two small but important new ones.

1) I still find that the language of the abstract is unclear and does not well communicate the directionality and significance of the findings. I would suggest the following text in place of lines 28-35:

“When calculated using NGHGI conventions, key global mitigation benchmarks become harder to achieve, requiring both earlier net zero CO₂ timing and lower cumulative emissions. Furthermore weakening natural carbon removal processes have the potential to mask anthropogenic land-based removal efforts, with the result that land-based carbon fluxes in NGHGIs may ultimately become sources of emissions by 2100. Our results are of critical importance to the Global Stocktake, suggesting that many nations will need to increase the ambition of their climate targets to remain consistent with global temperature goals.”

We think this is a very good suggestion and have updated the abstract accordingly.

2) I do agree with the authors’ argument that the term “indirect” is a reasonable one to apply to natural carbon sinks. However, my quibble is that only a very small portion of this flux is an indirect response to LULUCF itself — rather it is an indirect land carbon flux that is driven by mostly (>90%) fossil fuel emissions. So I still think the phrase “indirect LULUCF flux” implies causation from land management that is not the case.

3) Figure 1 — I like this new figure, but please do not use the term “virgin land” (this is quite a problematic term that carries misogynistic undertones) ... I would suggest maybe just “Other land” here instead.

We agree and have updated the language in the figure to “land with limited or no human activity”.

4) Finally it seems important to acknowledge either in Figure 1 or elsewhere in the manuscript that the terms “managed” and “unmanaged land” generally only refer to industrial management by mainstream economic activity. However, most of not all of what you refer to as “unmanaged” has in fact been sustainably managed by indigenous communities who have lived there for millennia.

We further clarify that while we used the term “unmanaged”, some of this land has been managed by indigenous communities for centuries to millennia in Figure 1’s legend.

Referee #3 (Remarks to the Author):

Thanks to the authors for taking serious time and effort to address all 3 (positive) reviews of this MS. The result is looking very good. It seemed pretty clear that most of the comments were around the presentation, and very little doubt that the science is robust and important. So I hope this can move towards acceptance and publication. I have a few comments though which hopefully will further improve clarity.

This is one of those issues which takes a while to fully comprehend. For me at least I have to think it through a couple of times to be clear. Therefore, it is very difficult for the authors, who are experts and already have this clearly in their mind, to see the stumbling blocks to clarity. Fig 1 really helps actually, but I found bits of the text still require having to stop and think through which way the “more” or “less” ambitious phrasing means.

For example, line 105; the “LULUCF convention results in more ambitious benchmarks...”. This could be read as “when countries use the LULUCF convention they set themselves targets/benchmarks which are more ambitious than if they had used a model-based convention” (i.e. they can actually relax their targets somewhat), or it could be read as “when countries rely on LULUCF convention to assess their progress towards benchmarks they need to make the benchmarks more ambitious” (i.e. they need to work harder to reduce emissions). The correct reading is the latter, but the reader shouldn’t have to stop and think it through.

We agree and have updated wording per the reviewer’s suggestion.

It might help to try to see this from the point of view of a new reader and spell it out much more simply than you might imagine is needed.

- The issue is that there are two ways of measuring land use emissions at national scale
 - If countries set a target based on modelling (e.g. as per IPCC timelines for the need for net-zero), but then monitor their progress based on accounting (which is what they should be reporting) they will miss their target (by emitting too much)
 - They therefore need to make the targets they set themselves more ambitious
 - o Emissions they report need to be lower in their reporting than the modelling would suggest
 - o Date of net zero they aim for needs to be earlier than modelling would suggest
- When this is implemented in the text it might not actually need much of a change. E.g. this sentence

(line 105):

“...LULUCF conventions results in more ambitious emissions benchmarks...”

could become:

“...LULUCF conventions results in the need to set more ambitious emissions benchmarks...”

Similarly, the sentence on line 109 that “emissions reductions ... are enhanced” could be mis-read as using LULUCF accounting is better because you end up with greater emissions reductions. Whereas it means that in order to meet targets, emissions reductions measured using LULUCF accounting must be greater than model-based numbers. So maybe change:

“Emissions reductions in 2030 relative to 2020 are enhanced by 3-6 percentage points for both pathway categories”

to

“Emissions reductions in 2030 relative to 2020 need to be 3-6 percentage points greater for both pathway categories if assessed using LULUCF accounting”

Apologies for a long-winded discussion on what will probably sound pedantic to you – but I think it will be vital for the success of the paper to be crystal clear to a new reader. As an aside, this whole issue is similar to the choice of CO₂-equivalence metric used to define net-zero GHG. IPCC WG1 has a horrible description (WG1 SPM, para D.1.8) of this which is so convoluted I don't think even the authors understand it 😊. It would be great if your paper can avoid also having technically-correct but near impossible-to-interpret text.

Regarding figure 1, my comment here might be problematic – maybe an issue the editor can contribute to – it feels like 1A is really useful up front as an explainer of the basic issue being addressed. But panel 1B feels like it should come after the results are shown – it almost pre-empts what the results are. In a perfect world I would recommend splitting the figure, and having 1B come after figure 2 at least. It would be useful during the discussion section to have 1B there. But I realise this increases the total number of figures, even if the total number of panels is unchanged. If the journal would be willing to accept that then it could help the flow. If that is not allowed, then I would be OK to keep them together as one figure.

We have split the old Figure 1 into its two panel components (now Figure 1 and Figure 4). We thought that it was best to move the second component after the discussion of shifting metrics to provide a more streamlined narrative for the reader.

Minor points

- Can you clarify what “land-based CDR” refers to (line 166) – from figure 4 it doesn't include BECCS right? So what you mean here is CDR which relies on land-based storage, rather than land-based removal from the atmosphere?

We have clarified the text here.

- Paragraph from line 181 – if you want a (quantitative) reference that future mitigation scenarios

reduce or even reverse the land-sink before the end of this century, then IPCC WG1 has a figure on this – chapter 4, fig 4.7 shows the land sink under the SSPs, and SSP1-1.9 sees a net negative “natural” land sink from around 2070s onwards. So from this point in a roughly 1.5-degree scenario the LULUCF accounting convention will actively work against countries ability to maintain net zero. This gives good consistency and support to your results in figure 2

We have added a reference here.

- make figure 2 the same colours as figure 1 – in 1A, green is NGHGI and red/blue are models, but in figure 2 these are swapped!

Great point, updated.

- Figure 3 – related to comments on text clarity, maybe the figure could have the zero line more prominent and label above it as “LULUCF-convention defines NZ earlier” for positive values, and “model conventions define NZ earlier” for negative values. Again, this makes it super clear and unambiguous for a reader not as familiar with it as the authors

We have confirmed that similar text is provided in the figure legend. We were not able to find a nice way to include annotations for this figure, but could discuss further if this is of interest to the editor.